# The effect of therapeutic potential and safety of bone marrow-derived against adipose-derived mesenchymal stem cells in aged mice associated with septic arthritis

**Alani Mohanad Khalid Ahmed** [1]*, **Mujahid Khalaf Ali**[1], **Basma Kh. Alani**[2]

1 Department of Microbiology, College of Medicine, Tikrit University, Salahaldin, Iraq, 2 Department of Biotechnology, College of Biotechnology, Al-Nahrain University, Baghdad, Iraq

* dr.mohaand@tu.edu.iq

## Abstract

Mesenchymal stem cells (MSCs) show potential for treating septic arthritis in aged populations, but their efficacy and safety in aged patients remain unclear. The objective of this study is to evaluate the therapeutic potential and safety profiles of bone marrow-derived MSCs (BM-MSCs) and adipose-derived MSCs (AD-MSCs) in aged murine model of septic arthritis. MSCs were isolated, characterized, and labeled for in vivo tracking. The experiment consisted of a total of 36 mice, which included 9 subgroups with four replicates per treated group: control group, treated groups (BM-MSC1, BM-MSC2, AD-MSC1, and AD-MSC2), and untreated groups (Un-BM1, Un-BM2, Un-AD1, and Un-AD2). The treated groups received MSC therapy following the induction of septic arthritis via intra-articular injection of *Staphylococcus aureus*. The results showed that BM-MSC1 significantly performed higher than AD-MSCs in reducing inflammation, promoting cartilage repair, and modulating immune responses. BM-MSC1 showed significant upregulation of regenerative markers such as interleukin-10 (IL-10) and collagen type II alpha 1 chain (COL2A1) and downregulation of pro-inflammatory markers such as tumor necrosis factor-alpha (TNF-a) and matrix metalloproteinase-13 (MMP-13). Imaging confirmed superior retention, engraftment, and host tissue interaction for BM-MSCs. AD-MSCs showed slightly lower efficacy and safety, highlighting the need for optimization. Untreated groups experienced severe inflammation, tissue degradation, and systemic organ damage, emphasizing the significance of intervention. The study had identified BM-MSC1 as a superior therapeutic option for septic arthritis in elderly populations and suggested AD-MSCs as an alternative in cases where extraction is not feasible. Future research can optimize MSC therapies, explore alternative sources, and conduct translational studies. Targeted preconditioning, combinatory approaches, and advanced molecular analyses are crucial for maximizing therapeutic outcomes.

**Data availability statement:** All relevant data are within the paper and its Supporting Information files.

**Funding:** The author(s) received no specific funding for this work.

**Competing interests:** The authors have declared that no competing interests exist.

## Introduction

Septic arthritis is a devastating and weakening condition characterized by the acute and often severe inflammation of joints due to bacterial infection [1]. Its progression leads to rapid joint destruction, cartilage degradation, and systemic complications, making it a clinical priority for effective and safe therapeutic interventions [2,3]. The challenge of addressing septic arthritis becomes particularly pronounced in aged populations, where immunosenescence and a diminished regenerative capacity exacerbate the impact of the condition. This dual vulnerability of age and disease pathology necessitates innovative strategies that not only target the infection but also restore joint integrity and functionality.

Mesenchymal stem cells (MSCs) have emerged as reliable candidates in the field of regenerative medicine due to their multipotent differentiation abilities, immunomodulatory properties, and paracrine effects [4]. Among the diverse sources of MSCs, bone marrow-derived MSCs (BM-MSCs) and adipose-derived MSCs (AD-MSCs) have gained significant attention. BM-MSCs is considered as a gold standard for stem cell therapies, possess robust osteogenic and chondrogenic potentials [5]. However, their accessibility is often limited by the invasive nature of bone marrow extraction and age-related decline in functionality. In contrast, AD-MSCs offer a less invasive alternative, which can be readily harvested from adipose tissue, present comparable immunomodulatory and regenerative properties [6]. Despite these advantages, the therapeutic efficacy and safety profiles of AD-MSCs relative to BM-MSCs remain contentious, particularly in the context of inflammatory conditions such as septic arthritis.

Aging adds another layer of complexity to this comparative analysis. As organisms age, the microenvironment of stem cell niches undergoes significant changes, influencing the regenerative capabilities of MSCs [3,7]. For instance, BM-MSCs from aged donors may display reduced proliferation and differentiation potential, while AD-MSCs may retain a more stable phenotype [8]. Moreover, the inflammatory milieu associated with septic arthritis can further modulate MSC behavior, altering their capacity to repair damaged tissue and regulate immune responses [9]. These dynamics underlay the importance of tailoring stem cell-based therapies to address the distinct challenges posed by aging and inflammation [10].

This study addresses several key problems including the limited understanding of how aging and septic arthritis influence the therapeutic potential and safety of BM-MSCs and AD-MSCs, lack of direct comparative data between these MSC sources in septic arthritis, and the scarcity of insights into their performance in aged hosts. The key objective of this work is to evaluate and compare the therapeutic efficacy and safety profiles of BM-MSCs and AD-MSCs in aged murine model of septic arthritis. It is hypothesized that AD-MSCs may demonstrate superior therapeutic potential in aged environments due to their resilience to age-related decline and robust immunomodulatory properties. The contributions of this research include advancing knowledge on the role of MSC source and host age in stem cell therapy outcomes, identifying optimal therapeutic approaches for elderly patients with septic arthritis, and informing future clinical strategies that integrate regenerative medicine with geriatric care.

## Materials and methods

### Animal model and ethical approval

Aged (22–24 months old) male and female C57BL/6 mice were utilized to replicate the physiological and immunological characteristics of elderly humans. The mice were raised according to standard protocol [11]. Ethical approval was obtained from Experimental Animal and Management Committee of Tikrit University (No.20241114-R89), ensuring compliance with guidelines for the animal use and humane treatment.

### Experimental design

A structured method was used to divide the mice into three main groups: a control group, treated groups, and untreated groups. The grouping strategy was used to subdivide the three main groups into a total of nine subgroups, including the control group, treated groups (BM-MSC1, BM-MSC2, AD-MSC1, and AD-MSC2), and untreated groups (Un-BM1, Un-BM2, Un-AD1, and Un-AD2). Un-BM1 and Un-BM2 represent untreated bone marrow-derived MSCs (type 1 and 2, defined in the next paragraph), while Un-AD1 and Un-AD2 represent untreated adipose-derived MSCs (type 1 and 2). Each group was replicated four times to reliably compare the control to treatments statistically thereby reducing the effect of random variability. Thus, the sample size in this study was 36 mice (18 males and 18 females), calculated using total sample size = number of groups × replicates per group × mice per replicate (sample size = 9 subgroups × 4 replicates per group = 36 mice). This is a deterministic computation method based on a fixed formula technique, which is commonly used in experimental design to ensure appropriate group allocation and statistical power.

The treated groups received MSC therapy (BM-MSC1, BM-MSC2, AD-MSC1, and AD-MSC2) after arthritis induction, while the untreated groups received no MSC therapy following arthritis induction and thus served as positive controls. The septic arthritis in the treated groups was induced by intra-articular injection of *Staphylococcus aureus*. A suspended *Staphylococcus aureus* ($10^7$ CFU in 10 μL of phosphate-buffered saline [PBS]) was injected into the right knee joint of each mouse. Careful monitoring of the mice to ensure the ethical application on model. On the other hand, untreated groups (Un-BM1, Un-BM2, Un-AD1, and Un-AD2) received PBS injected in the right knee joints. The only difference is that the control group received PBS without arthritis induction, while the untreated groups received PBS after arthritis induction. This method reliably replicates the clinical characteristics of bacterial joint infection, including inflammation, cartilage degradation, and immune dysregulation.

In this study, BM-MSC1 is a trabecular bone-derived MSC, which is a multipotent stem cell primarily isolated from the spongy part of bones for its enhanced osteogenic differentiation, while BM-MSC2 is an endosteal niche MSC, which is a bone marrow cell derived from the endosteal region that plays a role in bone remodeling and hematopoiesis. Furthermore, AD-MSC1 is an infrapatellar fat pad MSC isolated from knee joint fat pads, which is generally used in cartilage repair because of its chondrogenic potential, while AD-MSC2 is a visceral AD-MSC present in deeper fat layers, which can differentiate into bone and cartilage cells.

The flow cytometry was used to differentiate between type 1 and type 2 MSCs for the groups based on their origin, functional properties, and marker expression. Type 1 MSCs show higher expression of CD73, CD90, and CD105, while type 2 MSCs show higher expression of CD44, CD146, and growth factors (VEGF, etc.). BM-MSC1 are derived from bone marrow and have strong anti-inflammatory and immunomodulatory properties but differentiate into osteoblasts and chondrocytes but with slower proliferation rates. BM-MSC2 increased expression of VEGF, hepatocyte growth factor (HGF), and FGF. AD-MSC1 had excellent immunosuppression.

### Methods of alleviate suffering and sacrifice

The research rats were provided with NSAIDs such as carprofen (5 mg/kg oral [PO], every 24 hours) to produce efficient analgesia and reduce inflammation during the experimental periods. The carprofen was used to aid in the management

of mild to moderate pain, especially following the incision process. It was administered via the feed to minimize gastrointestinal adverse effects, and the rats were monitored for negative reactions or signs of discomfort. In accordance with the institutional committee of the standards for humane care at Tikrit University, the right dosage and duration are determined by the procedure and the state of the animal.

The humane euthanasia method was used in sacrificing the rats. Research rats were carefully handled to minimize pain and stress. The preemptive analgesia was administered before any sacrifice was performed to increase the humane process. The rats were sedated via the pentobarbital method using euthanasia solution containing 100 mg/mL of sodium pentobarbital. The dosage (between 100 and 150 mg/kg) was determined based on the rat's weight. The dosage was administered via intravenous (IV) injection while carefully restraining the rat, using a 25–30G needle to ensure venous access via the lateral tail vein, and the injection was slowly delivered to avoid extravasation. All vital signs, such as cessation of respiration, heart rate, cardiac arrest, and reflexes, were monitored. Confirm death using a secondary method, such as cervical dislocation, before samples are taken. The anesthetic depth was maintained during the operation, using warmers to prevent hypothermia, to ensure that the righting reflex disappeared and with no reaction to a toe pinch.

### Isolation and characterization of MSCs

A second generation (cultured MSCs) was used in this study. MSCs were derived from two sources, the BM-MSCs and AD-MSCs. Firstly, BM-MSCs were harvested from the femurs and tibiae of the donor (Fig 1) using standard bone marrow aspiration and biopsy techniques, followed by density-gradient centrifugation to isolate the MSC population. Separate donor mice were used consisting of pooled BM-MSCs from five healthy donor mice were cultured and expanded and used only for BM-MSC treatment group, while pooled AD-MSCs from four healthy donor mice were cultured and expanded and used only for AD-MSC treatment group. Pooling cells from multiple donors, a common practice to reduce biological variability. The MSC yield per donor mouse was ~0.5–2 million cells/mouse (after culture expansion to passage 2–3) for BM-MSCs and ~2–5 million cells/mouse for AD-MSCs. The intra-articular injection of MSCs in mice was performed by first resuspend $1 \times 10^6$ MSCs in 15 µL of sterile PBS or saline, keeping the suspension cold and sterile. The mice were anesthetized using xylazine and confirm deep anesthesia before placing it in a supine position. The knee area was sterilized with 70% ethanol, then extend the leg to locate the patellar tendon above the tibial tuberosity. A needle (30–32 gauge insulin syringe) was inserted at a ~45° angle under the patella into the joint cavity and slowly injected the MSCs (10 µL). After injection, the mice were monitored until they recovered (for joint swelling or distress. Secondly, AD-MSCs were extracted from subcutaneous adipose tissue through enzymatic digestion with collagenase, and the resulting cell suspension was purified by filtration and centrifugation. Cells were cultured and labeled with fluorescent dyes (PKH26 and CFSE) for in vivo tracking.

The methods of isolation and characterization of MSCs were adapted from [12,13]. Falcon Cell Strainers (100 µm) were used to filter the bone marrow aspirate to remove coagulated debris, with a minimal loss of volume (< 5%). A growth-promoting medium for MSCs comprising minimal essential medium (MEM)-α, 10% MSC-grade Biowest Foetal Bovine Serum (FBS), glutamax, and 10 µg/mL gentamicin was then used to dilute the filtered aspirate at a 1:4 ratio. The medium was withdrawn after two days of incubation at 37°C with 5% $CO_2$, and adherent cells were then washing five times using PBS. The successful extraction of red blood cells was confirmed by microscopy. Five more days were subsequently spent cultivating MSC colonies. After developing for seven days in primary tissue culture flasks, cells were trypsinized and replated to produce a homogeneous cell population. The initial trypsinization was carried out at this seven-day point to ensure a substantial cell growth while avoiding colony overlap. Confluence of cells was obtained between 65% to 75%. Cells from the three groups used in this study show satisfactory growth rates. The colonies that developed following the initial trypsinization were referred to as Passage Zero. Following confluence, the cells in these flasks underwent trypsinization and were cryopreserved in cryovials for potential future growth utilizing quantum flex cell expansion system (Terumo BCT). These cells were identified as expansion passage 1 (Fig 1). Cells from all three experimental passages (1–3) were imaged on a phase contrast microscope using a 40× objective with a 10x eyepiece (Nikon Eclipse Ts2).

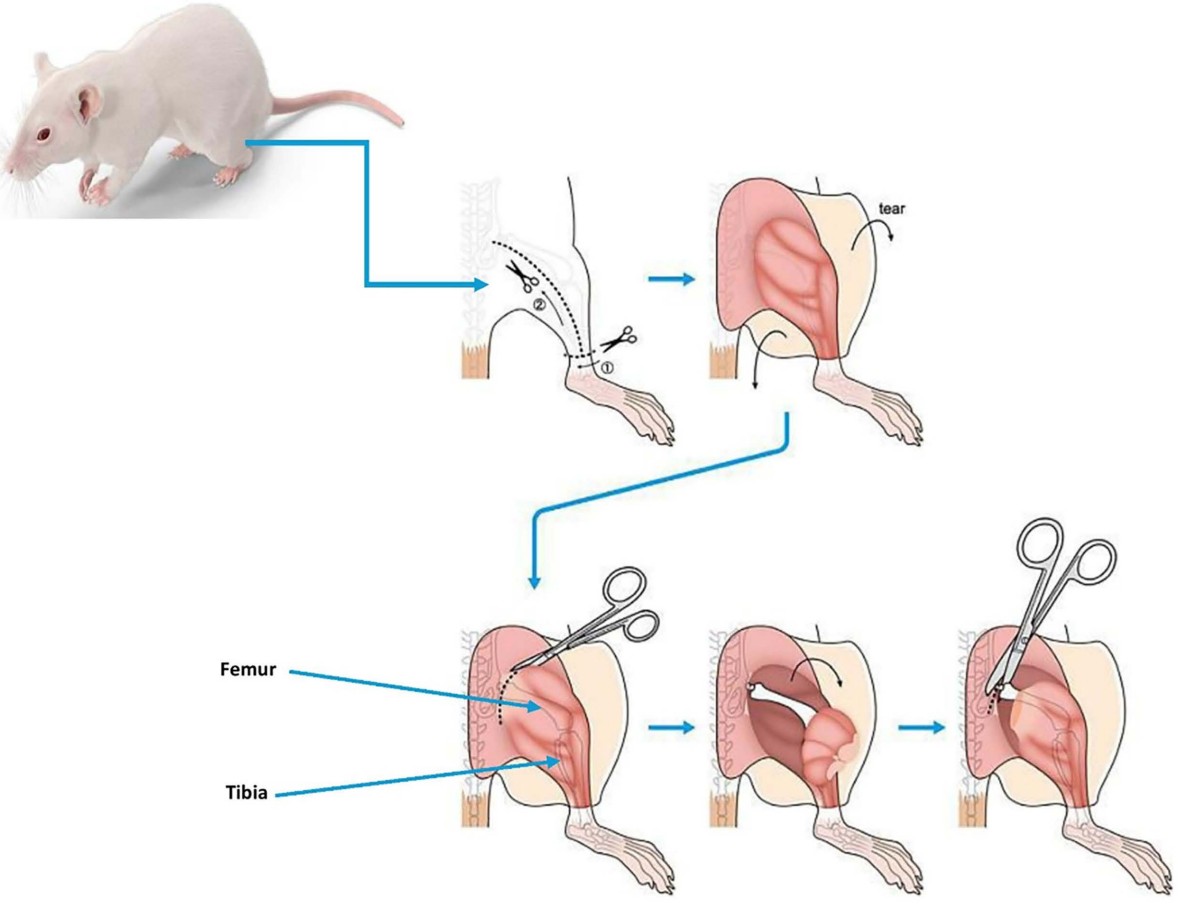

**Fig 1. Femurs and tibiae of donor mice using standard dissection technique.** A total of 36 samples were collected from 36 mice: 9 subgroups with four replicates per treated group: control group, treated groups (BM-MSC1, BM-MSC2, AD-MSC1, and AD-MSC2), and untreated groups (Un-BM1, Un-BM2, Un-AD1, and Un-AD2).

The International Society for Cellular Therapy (ISCT) criteria for MSCs were used in this study, which were based on the following: (i) adherence to plastic surfaces, allowing their isolation and expansion; (ii) specific surface marker expression, for which MSCs must express CD73, CD90, and CD105 while lacking hematopoietic markers such as CD34, CD45, etc; and (iii) differentiation potential, requiring the capability to differentiate into osteogenic (bone), adipogenic (fat), and chondrogenic (cartilage) lineages under specific conditions. Both BM-MSCs and AD-MSCs are cultured in similar conditions using MEM-α, with low glucose (1 g/L) for BM-MSCs and high glucose (4.5 g/L) for AD-MSCs, but supplemented with 15% MSC-competent FBS, 2 mM L-glutamine, and antibiotics (Gentamicin, Sigma-Aldrich). Cells were incubated at 37°C with 5% $CO_2$ and 95% humidity, with BM-MSCs seeded at 3,000–6,000 cells/cm² and AD-MSCs at 5,000–10,000 cells/cm². Passaging occurred every 5–7 days at 60–80% confluence utilizing Trypsin-EDTA, maintaining cultures within this period to prevent senescence. MSC identity is confirmed through surface marker analysis (CD73+, CD90+, CD105+, CD34-, CD45-) and differentiation assays for osteogenic, adipogenic, and chondrogenic potential. Alizarin Red S staining was performed after 21 days for osteogenic differentiation.

## Differentiation protocol

To initiate osteogenic differentiation, the MSCs were first seeded at a high density of $1 \times 10^5$ cells per well in fibronectin-coated 6-well plates and cultured until they reached 80–90% confluency [13,14]. The growth medium was then replaced with an osteogenic induction medium containing 0.1 μM dexamethasone, 10 mM β-glycerophosphate, and 0.2 mM ascorbic acid-2-phosphate, which was refreshed every 2–3 days over a 21-day culture period until mineralized bone nodules formed [13,14]. To confirm and quantify the calcium deposition, Alizarin Red S staining was performed by first fixing the cells in iced-cold 70% ethanol for one hour, rinsing twice with distilled water, and then incubating with a 2% Alizarin Red S solution (pH 4.1–4.3) for 30–45 minutes [14]. After thorough washing (five times) to remove non-specific stain, the bound dye was eluted with 10% cetylpyridinium chloride, and the absorbance of the eluate was measured at 562 nm for quantification [14].

To induce adipogenic differentiation, MSCs were first plated at a high density of $1 \times 10^6$ cells in a 10 cm plate and cultured until they reached 100% confluency, forming a complete monolayer [15,16]. The growth medium was then replaced with an adipogenic induction medium containing a hormonal cocktail, which included indomethacin to trigger lipid accumulation, and this medium was replaced every 3–4 days [16]. After a culture period of 21 days, mature adipocytes displaying large intracellular lipid vacuoles were confirmed using Oil Red O staining, which was performed by first washing the cells with PBS, fixing them with 4% formaldehyde for 15 minutes, and then incubating them with a freshly prepared and filtered 0.2% Oil Red O working solution for 30 minutes at room temperature. Following thorough washing with distilled water to remove unbound dye, the bound dye was eluted using 100% isopropanol and quantified by measuring the absorbance of the eluate photometrically at 510 nm.

To induce chondrogenic differentiation, MSCs were plated at a density of $2 \times 10^5$ cells in a 96-well U-bottom suspension culture plate to allow for spontaneous spheroid formation within 24–48 hours, after which they were induced with a chondrogenic medium containing TGF-β, dexamethasone, ascorbate, and proline, and the medium changed every 3 days for 21 days to promote the production of a proteoglycan-rich extracellular matrix [15,17]. To confirm successful differentiation via sulfated glycosaminoglycan detection, the spheroids were collected, washed with PBS, and fixed in 4% paraformaldehyde for 2 hours before subjected to staining with a 1% Alcian Blue solution in 3% glacial acetic acid (pH 2.5) for 30 minutes to 24 hours. After thorough washing with distilled water to reveal a dark blue-stained matrix, the spheroids were dehydrated, embedded in paraffin, sectioned, and mounted on slides for final histological analysis.

## Staining protocols

The Hematoxylin and Eosin (H&E) staining to view the cellular morphology was performed, beginning with deparaffinized and rehydrated tissue sections [18]. The slides were immersed in Mayer's Hematoxylin solution for 3–10 minutes to stain cell nuclei a deep blue-purple, then rinsed under running tap water for 5–10 minutes to remove excess stain. Next, the slides were counterstain by immersing in Eosin Y solution for 1–5 minutes. Lastly, the sections were dehydrated through a graded series of alcohols, clear them with xylene, and mounted in synthetic resinous mounting medium, allowing for further assessment.

The Safranin-O (Saf-O) staining was caried out by preparing paraffin sections. After deparaffinization and rehydration, the nuclei was stained with Weigert's Iron Hematoxylin for about 5–10 minutes, then washed the slides thoroughly under running tap water [19]. The sections were differentiated in a fast green solution for 1–3 minutes and counterstain was then done by rinsing in 1% acetic acid to prevent carry-over before applying 0.1% Saf-O solution for 5–30 minutes. After a final rapid dehydration through graded alcohols, clearing in xylene, and mounting, the proteoglycan-rich cartilage matrix was stained, providing a clear visual and semi-quantitative measure of GAG content.

The Masson's Trichrome staining was performed, with the process begins with standard deparaffinized and rehydrated sections [20]. The nuclei were first stained with a regressive hematoxylin solution for 5–10 minutes, followed by washing and differentiation. The slides were then treated with Biebrich Scarlet-Acid Fuchsin solution for 5–15 minutes to stain

cytoplasm and muscle fibers red, followed by differentiation in a phosphomolybdic/phosphotungstic acid solution until collagen was decolorized. Lastly, the sections were transferred directly to aniline blue or light green solution for 1–5 minutes to stain the collagen fibers blue or green, before rinsed, dehydrated, cleared, and mounted.

The immunohistochemical staining (IHC) of COL2A1 (Type II Collagen) was used as a definitive marker for chondrocytes and mature cartilage [21]. After deparaffinization and rehydration, performed heat-induced epitope retrieval by incubating the slides in a citrate-based buffer (pH 6.0) in a water bath for 20–40 minutes, cooled and then blocked endogenous peroxidase activity with 3% hydrogen peroxide. Block non-specific sites with a normal serum for 20–30 minutes before incubating the sections with a primary monoclonal antibody against COL2A1 overnight at 4°C. The next day, the bound antibody was visualized using a labeled streptavidin-biotin (LSAB), which produced a brown precipitate at the site of type II collagen localization, followed by a light hematoxylin counterstain to visualize nuclei.

The immunohistochemical staining of SOX9, a marker for chondroprogenitor/cells, was conducted [22]. Begin with deparaffinized and rehydrated sections and perform heat-induced epitope retrieval using a high-pH EDTA or Tris-EDTA buffer (pH 9.0) for 20–30 minutes to expose the nuclear epitopes. After cooled and blocked peroxidase activity, a protein block applied to reduce non-specific background before incubating with a specific anti-SOX9 primary antibody for 60–90 minutes aovernight at 4°C. The detection is then carried out using a polymer based HRP detection system with DAB as the chromogen, resulting in a brown staining localized, which was then lightly counterstained with hematoxylin, allows for the identification and quantification of chondrogenic progenitor cells within the tissue.

## Cell labeling and tracking

MSCs were labeled for in vivo tracking utilizing PKH26 (lipophilic membrane dye) and CFSE (cytoplasmic dye) before injection, allowing appropriate monitoring of their localization and behavior within the target tissues [23–25]. MSCs (1–5 × 10$^6$ cells/mL) for PKH26 labeling were performed by centrifuging, resuspending in Diluent C, and incubating with 4 μL PKH26 (2 μM) for 5 minutes at 22–24ºC, then quenching with 2 mL FBS. Subsequently, triple PBS washes and resuspension for injection were conducted. Fluorescence microscopy was used to identify cells that have been PKH26-labeled (551 nm excitation, and 567 nm emission). CFSE labeling was performed by suspending MSCs in PBS, incubated with 4–10 μM CFSE for 10 minutes at 37°C in the dark, quenching with FBS, washed, and resuspended at 1 × 10$^6$ cells/100 μL for the injection. Both dyes enable real-time tracking via microscopy. The labeling process was carefully optimized to ensure that the dyes were effectively integrated without affecting cell viability or function. Confocal microscopy was employed to visualize the labeled cells at high resolution, confirming successful dye uptake and uniform distribution, while flow cytometry provided quantitative data on labeling efficiency and the preservation of cellular integrity.

## Therapeutic administration

Within 24 hours after infection, the intra-articular injection of 1 × 10$^6$ MSCs (BM-MSCs and AD-MSCs) suspended in 10–20 μL of sterile PBS was given to the injured knee joint, targeting the acute inflammatory phase [26,27]. A 30G needle was used for the injection to assure accurate delivery into the joint region while under anesthesia. Mice in the control group received PBS without arthritis induction, while the untreated groups received PBS after arthritis induction. The intervention in model mice aimed to explore the immunomodulatory and regenerative properties of MSCs in mitigating inflammation and promoting joint repair.

To confirm the development of arthritis, firstly, we used intra-articular injection of monosodium iodoacetate (MIA)-induced osteoarthritis model, which was established to triggers a rapid and massive inflammatory response within hours, characterized by the release of proteases and inflammatory cytokines (IL-1β, TNF-α) [28–30]. Secondly, we run a small pilot study to characterize our model by induced arthritis in a separate group of mice (n = 5) and samples were taken at various points (i.e., 6h, 24h, 3d, and 7d) and confirm histologically, though the cartilage degradation was not yet visible. Thus, arthritis was confirmed to be successfully induced based primarily on observed joint swelling and pain behaviors at

that time, with the rationale for the early therapeutic intervention based on well-established understanding that the critical early inflammatory phase is the prime target for MSC therapy. The definitive proof of established arthritis (cartilage erosion, bone changes) would come from the final histological analysis at the end of the study, comparing the treated groups to the untreated arthritic controls.

## Outcome assessment

Many important outcomes, with an emphasis on inflammation, pain, tissue health, and immunological response, were evaluated to fully assess the efficacy of MSC therapy. Joint swelling as a visible marker of inflammation was measured precisely using calipers, while weight-bearing tests offered understandings into pain levels and functional recovery. To examine the structural integrity of the joint, histological analyses of tissue sections were conducted using Hematoxylin and Eosin (H&E), Saf-O staining, Masson's trichrome, COL2A1, and SOX9 to reveal detailed patterns of cartilage degradation, bone erosion, and synovial inflammation. Immune responses were closely monitored by measuring key cytokines, including TNF-α, IL-6, and IL-10, in both synovial fluid and serum using ELISA (Thermo Fisher Scientific Inc), shedding light on both local and systemic inflammatory changes. Additionally, flow cytometry was employed to identify and quantify immune cell populations, such as regulatory T cells (Tregs) and macrophages in the joint environment.

## Safety evaluation

To ensure the safety of MSC therapy, a comprehensive evaluation was conducted to monitor potential adverse effects, focusing on both systemic and localized parameters. Liver and kidney function tests, including the measurement of alanine aminotransferase (ALT), aspartate aminotransferase (AST), creatinine, and urea levels, were performed to detect any signs of toxicity or organ dysfunction resulting from the treatment. These biomarkers provided a detailed overview of the metabolic and excretory systems, which are often sensitive to systemic disruptions.

## Comparative analysis

The therapeutic outcomes of BM-MSCs and AD-MSCs were carefully compared to identify differences in their efficacy and safety profiles. Key parameters, including the resolution of inflammation, the extent of cartilage repair, and the modulation of cytokine levels, were evaluated to determine their relative therapeutic impact. Advanced statistical analyses, such as two-way ANOVA, were employed to assess the significance of observed differences, ensuring robust and reliable conclusions. This comparative approach provided understandings into the distinctive biological mechanisms and therapeutic potential of each MSC type, while also identifying any variations in safety or off-target effects.

## Engraftment imaging

MSCs were subjected to RNA sequencing both prior to and following delivery in order to examine dynamic shifts in the expression of genes linked to inflammation and aging. This approach aided in the identification of important genes and pathways that underpin the therapeutic activities of MSCs by providing a comprehensive understanding on their adapt to host milieu. Total RNA was extracted using Sigma-Aldrich® TRIzol™ method kits. The quantity and quality of RNA were measured using Nanodrop to ensure that the extracted RNA was unaffected. The FastQC program was used to perform quality checks on raw sequencing reads, and Trim Galore trimming software was used to eliminate adapters and low-quality reads in order to ensure accurate and clean results. Data analysis includes normalizing the data, measuring gene expression levels, and mapping sequencing reads to a reference genome. Key gene expression changes were validated by qPCR using cDNA synthesized from total RNA utilizing the iScript cDNA Synthesis Kit. SYBR Green Master Mix was used for the qPCR reactions on a Bio-Rad CFX96 or Applied Biosystems StepOnePlus system, which ensured accurate amplification and detection. Specific primers were designed for target genes, including p16 (CDKN2A) as a senescence

marker (F: 5'-GCTGCCCAACGCACCGAATAG-3', R: 5'-CCGCCCGTCCCCTTGAATC-3'), IL-6 as an inflammation marker (F: 5'-ACTCACCTCTTCAGAACGAATTG-3', R: 5'-CCATCTTTGGAAGGTTCAGGTTG-3'), and GAPDH as a house-keeping gene for normalization (F: 5'-GAAGGTGAAGGTCGGAGT-3', R: 5'-GAAGATGGTGATGGGATTTC-3'). Reliable confirmation of the differential gene expression found in RNA sequencing was obtained by this qPCR validation, for the importance of inflammatory and aging pathways linked to MSCs.

In addition to this, advanced imaging techniques such as multiplex imaging and immunohistochemistry (Lunaphore COMET™) were utilized to visualize the engraftment of MSCs and their interactions with host tissues at the cellular level. These methods allowed for precise localization of the MSCs within the affected areas and offered key insights into their integration, retention, and functional contributions to tissue repair and immune regulation.

## Statistical methods

This study employs many statistical methods to ensure strong and reliable outcomes. The RNA sequencing was carried out for the differential gene expression analysis using DESeq2 and EdgeR. Normalization was done to prevent batch effects and ensure precise quantification. The Wald test (DESeq2) and the exact test (EdgeR) were employed to test for statistical significance. Multiple comparisons were controlled for using the false discovery rate (FDR) correction (Benjamini-Hochberg). A criterion of adjusted p-value (q) < 0.05 and $|\log_2$ fold change$| > 1.5$ was chosen to identify significantly differentially expressed genes, with 95% confidence intervals (CI) determined for fold changes. Pathway enrichment analysis uses Fisher's Exact Test, with significance assessed at FDR < 0.05, using Enrichr and KEGG.

Normalized to GAPDH as a housekeeping gene, gene expression was examined using the ΔΔCt method for qPCR validation. GraphPad Prism and Bio-Rad CFX Manager Software were used to statistically analyse the data. One-Way ANOVA (for multiple groups) and Tukey's post hoc test (for multiple comparisons) were used to compare the gene expression levels of the MSC-treated, untreated, and control groups. 95% CIs were computed for mean expression levels, and a significance level of α = 0.05 was used. Fold change thresholds for statistical significance were either >2 or <0.5 with p < 0.05.

Statistical comparisons between treatment and control groups were conducted for the in vivo MSC study using one-way ANOVA for multiple groups. ImageJ software was used to quantify imaging data, and histological scoring and cytokine level differences were statistically evaluated using two-way ANOVA, and then GraphPad Prism was utilized for statistical analysis. For histological scoring, the Bone and Cartilage Repair Scoring (Modified Pritzker Score): 0–5 scale: Normal cartilage (0) to complete cartilage erosion (5). This scoring method is commonly used in animal models to assess cartilage damage and subchondral bone changes.

All in vivo analyses were performed at a significance level of 0.05, and 95% confidence intervals were reported for histology scores, inflammatory indicators, and other outcome variables.

## Results

### Isolation and characterization of MSCs

Fig 2 presents the isolated MSC morphology and characterization. The MSC cultured in medium exhibited a fibroblast-like structure with multi-differentiation characteristics, which revealed their regular and directional nature. MSC were characterized by surface marker expression using FACS testing. The results indicated that the MSC expressed CD73+, CD90+, CD105+, CD45−, and CD34− (Fig 2A). The CD45−, and CD34− were hematopoietic lineage markers, which agreed with previous reported MSC characteristics [4,6,13]. Based on this result, a differentiation assay was utilized to assess the multi-differentiation potential of the MSCs. Significant morphological changes with lipid vacuole development were noted following 14 days of induction toward the adipogenic potential. Alizarin Red S staining after 21 days of osteogenic differentiation confirmed that MSC had differentiated into osteocytes (Fig 2B). However, the cells did not express CD14/CD11b, CD19/CD79a, and HLA-DR surface markers.

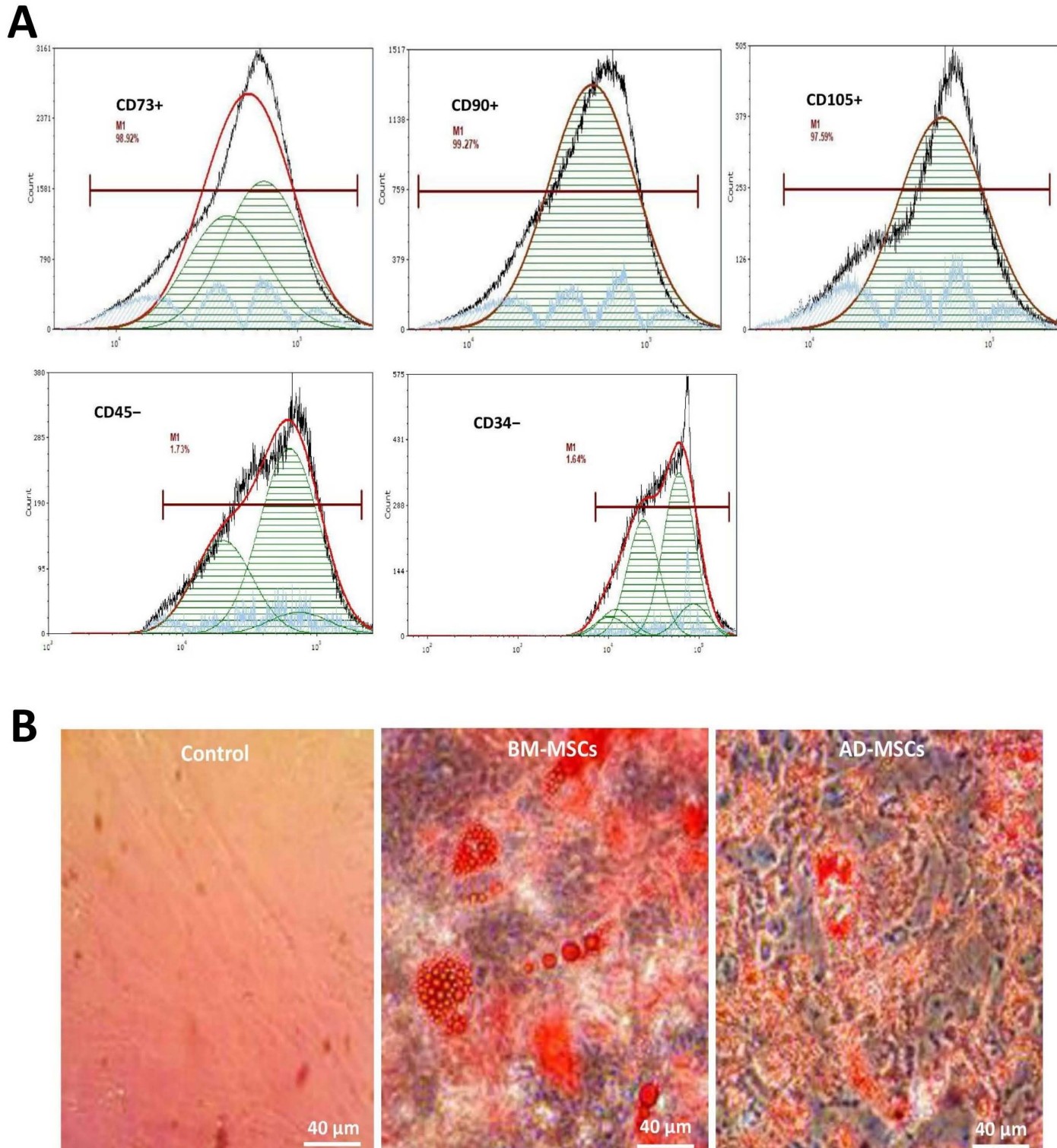

**Fig 2. MSC characterization.** (A) Flow cytometry showed that MSC were uniformly positive for CD73 +, CD90 +, and CD105 + expression (images in first row), while uniformly negative for CD45− and CD34 − expression (images in second row). (B) Isolated MSC appeared as fibroblast-like structures with multi-differentiation characteristics. Mice used was n = 9.

Table 1 presents the result of mesenchymal phenotype. Positive Markers (CD73$^+$, CD90$^+$, CD105$^+$) with high expression percentages confirm the cells display the classic mesenchymal phenotype, while the negative Markers (CD45$^-$, CD34$^-$) displays the absence of hematopoietic lineage markers validates the mesenchymal origin of the cells. This data indicates that the isolated cells meet the standard criteria for mesenchymal stem cells as defined by the International Society for Cellular Therapy (ISCT).

MSCs from the three distinct groups (control, BM-MSCs, and AD-MSCs) were cultivated and examined through three passages in order to describe cell morphology (Fig 3). The control group (column 1) had a fibroblastic look with bigger and stretched out structure, multipolar cell bodies and acquired similarity like spindle-shaped cells as they approached convergence. The distinctive fibroblast-like structure of BM-MSCs (column 2) was characterized by their elongated, multipolar cell bodies, which grew increasingly compacted honeycomb like structure as the cells approached convergence. Many of the BM-MSCs had noticeably relatively moderate nuclei, which was a distinguishing characteristic. In contrast to the two previous groups, AD-MSCs (column 3) differed significantly, exhibiting an epithelial-like structure with less compact cell bodies, less uniform diameters, and flattened cells with polygonal shapes compared to BM-MSCs.

## Cell labeling and tracking

Table 2 displays cell labeling and tracking metrics. The fluorescence intensity values, ranging between 91% and 95%, demonstrate high and consistent labeling efficiency of the fluorescent dyes, with minimal variability (±2%) indicating a reliable and reproducible labeling procedure. Cell viability was high (> 88.0), confirms that the labeling process does not significantly affect the functions or health of the MSCs. Fig 4 shows a steady and predictable increase, showing consistency in the results. These findings highlight the compatibility of the labeling protocol with preserving MSC integrity, ensuring their efficacy for accurate in vivo tracking and therapeutic applications. Altogether, the results validate a robust and efficient labeling strategy that achieves both effective fluorescence tagging and excellent cell viability, with minimal variability, supporting the credibility and reproducibility of the experimental method.

Fig 5 illustrates the mean therapeutic administration success rates across different groups, showing distinct trends between treated and untreated groups. The control group had a moderate success rate at 91.5%. Treated groups, including BM-MSC1, BM-MSC2, AD-MSC1, and AD-MSC2, demonstrate increased trend in success rates, with BM-MSC1 achieving the highest success rate at 93.5%, followed closely by BM-MSC2 (93%), AD-MSC1 (92.5%), and AD-MSC2 (92%). Conversely, untreated groups, including Un-BM1, Un-BM2, Un-AD1, and Un-AD2, show a progressive decline in success rates, with Un-BM1 at 90.5% while Un-AD2 reaching the lowest at 89%. This pattern suggests that therapeutic treatments significantly increase success rates compared to untreated conditions, with BM-MSCs showing higher efficacy than AD-MSCs.

## Outcome assessment of inflammation and pain

Fig 6 illustrates the outcomes of inflammation and pain assessment across different groups, showing significant improvements in the treated groups compared to the untreated groups. Treated groups show a clear trend of enhanced

**Table 1. Result of mesenchymal phenotype.**

| Marker | Expression | Percentage of Cells Expressing Marker | Description |
|---|---|---|---|
| CD73 | Positive (CD73$^+$) | 95% | Highly expressed on MSCs; key in cellular signaling. |
| CD90 | Positive (CD90$^+$) | 98% | Common MSC marker; important for adhesion and immunomodulation. |
| CD105 | Positive (CD105$^+$) | 92% | A core MSC marker associated with vascular interactions. |
| CD45 | Negative (CD45$^-$) | 0% | Absence confirms non-hematopoietic lineage. |
| CD34 | Negative (CD34$^-$) | 0% | Typically absent in MSCs, differentiating from hematopoietic stem cells. |

**Note:** Mice used n = 12.

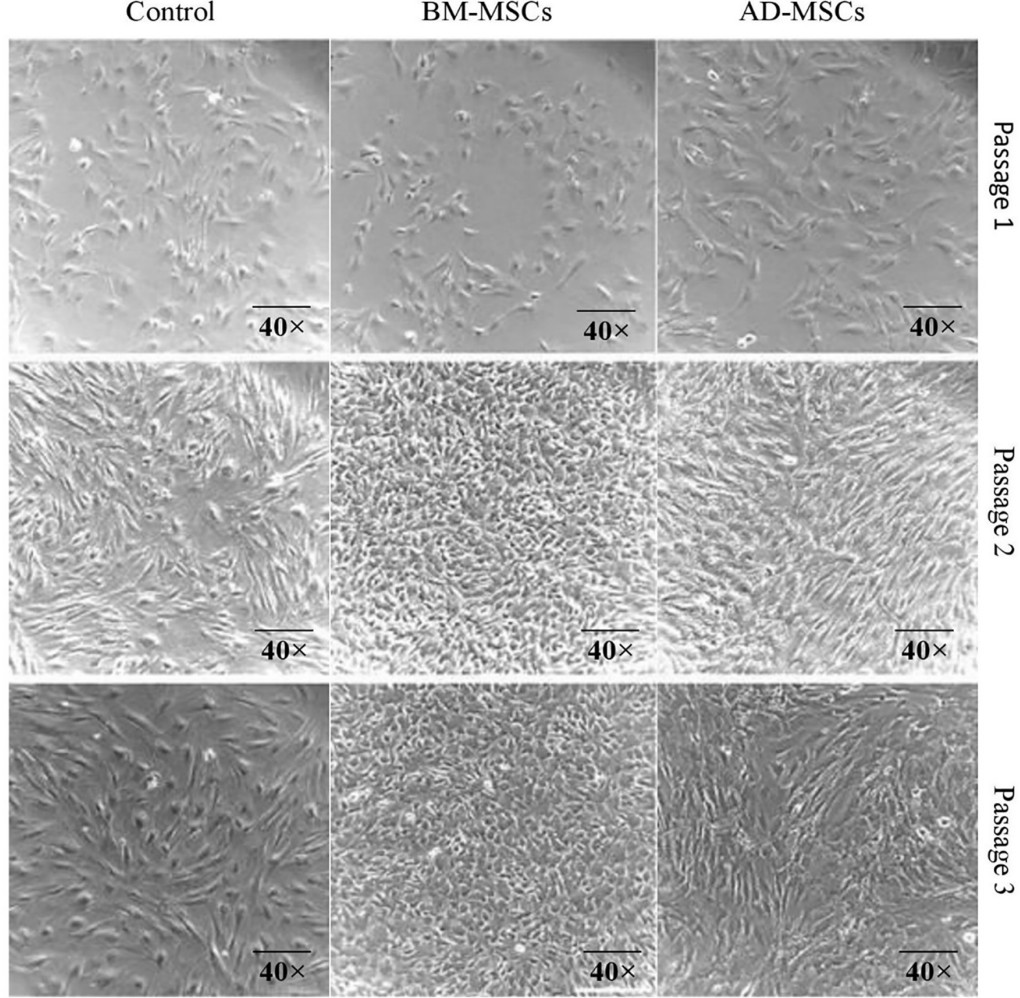

**Fig 3. Morphologic map of the MSC differences between the three groups.** A phase contrast microscope (40×) images of the cells across three passages were captured to feature the morphology of cells over time. Rows indicate passage number for each cell, while columns indicate group characteristics. Scale bar presents l = 220 μm. Mice used was n = 12.

therapeutic efficacy, with greater swelling reduction (ranging from 2.0 to 2.5 mm) and improved weight-bearing capacity (25–30%). Among these, BM-MSC1 has the highest efficacy, achieving 2.5 mm swelling reduction and 30% weight-bearing improvement, followed closely by BM-MSC2. Even though AD-MSCs also show positive effects, they are less effective than BM-MSCs. In contrast, untreated groups (Un-BM1, Un-BM2, Un-AD1, Un-AD2) display minimal therapeutic benefits, with swelling reduction limited to 1.0–1.3 mm and weight-bearing improvement as low as 9–12%.

ALT and AST levels are elevated in all treated groups compared to the control, with the highest levels observed in untreated groups, suggesting potential liver stress or injury. In contrast, BM-MSC and AD-MSC-treated groups show moderate increases in ALT and AST levels, indicating a more controlled systemic response and a safer liver profile compared to the untreated groups. These results emphasize the superior potential of treated interventions, particularly BM-MSCs, in mitigating joint inflammation and improving pain-related functions.

Table 3 presents the results of pain and inflammation expressions. The findings show that the treated groups (BM-MSC1, BM-MSC2, AD-MSC1, and AD-MSC2) outperformed the untreated groups (Un-BM1, Un-BM2, Un-AD1, and

**Table 2. Mean Cell Labeling and Tracking Metrics.**

| Group | Mean Fluorescence Intensity (%) | Mean Cell Viability (%) |
|---|---|---|
| Control | 97.5** | 97.5** |
| BM-MSC1 | 94.0** | 95.7** |
| BM-MSC2 | 93.5** | 94.7** |
| AD-MSC1 | 92.3* | 94.0** |
| AD-MSC2 | 91.5* | 93.0** |
| Un-BM1 | 92.3* | 91.5* |
| Un-BM2 | 91.5* | 91.0* |
| Un-AD1 | 90.5* | 89.5* |
| Un-AD2 | 89.5* | 88.5* |

Note: BM-MSC1, BM-MSC2, AD-MSC1, AD-MSC2 representing treated groups, while Un-BM1, Un-BM2, Un-AD1, Un-AD2 represent untreated groups. Mesenchymal stem cells (MSCs); bone marrow-derived mesenchymal stem cells (BM-MSCs) and adipose-derived mesenchymal stem cells (AD-MSCs). Mice used $n = 12$. One-Way ANOVA with statistical significance at $p < 0.05$ (*$p < 0.05$, **$p < 0.01$).

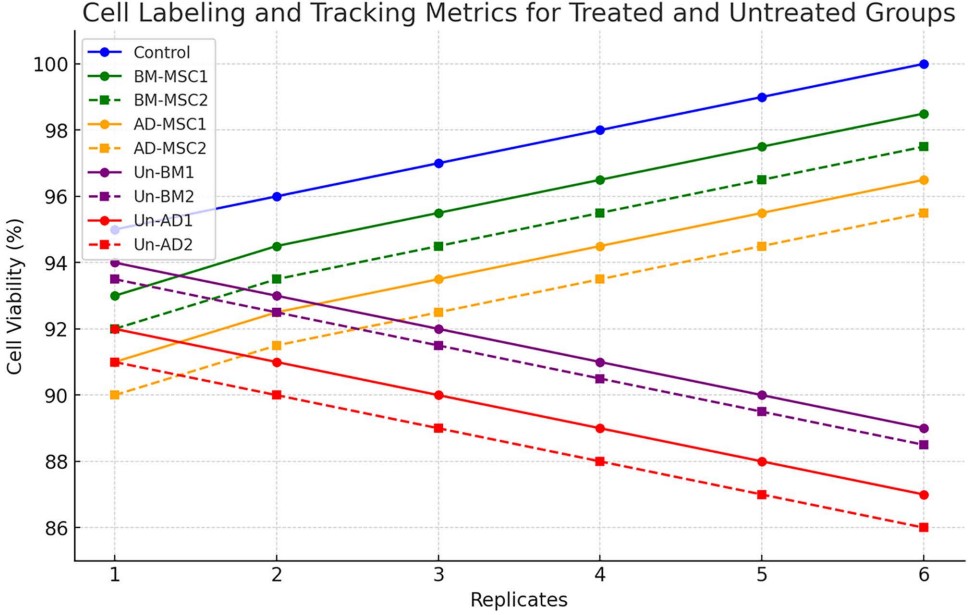

**Fig 4. Chart illustrates the treated groups (BM-MSC1, BM-MSC2, AD-MSC1, AD-MSC2) increasing in cell viability over replicates, while the untreated groups (Un-BM1, Un-BM2, Un-AD1, Un-AD2) show a gradual decrease.** Mesenchymal stem cells (MSCs); bone marrow-derived mesenchymal stem cells (BM-MSCs) and adipose-derived mesenchymal stem cells (AD-MSCs). Mice used $n = 12$. One-Way ANOVA with statistical significance at $p < 0.05$. (*$p < 0.05$, **$p < 0.01$).

Un-AD2) in terms of lowering pain and inflammation. Joint swelling and mild inflammation scores were lower in treated groups compared to moderate to severe inflammation scores and higher swelling in untreated groups. The treated groups had greater pain sensitivity thresholds, which indicated less pain, while the untreated groups had lesser thresholds, which indicated higher pain levels. TNF-α and IL-6 as two pro-inflammatory cytokines were considerably lower in treated groups, especially in BM-MSC1 (24 pg/mL IL-6 and 19 pg/mL TNF-α), whereas untreated groups had higher levels. Moreover, treated groups showed significantly lower levels of pro-inflammatory gene expression (a fold change) than untreated

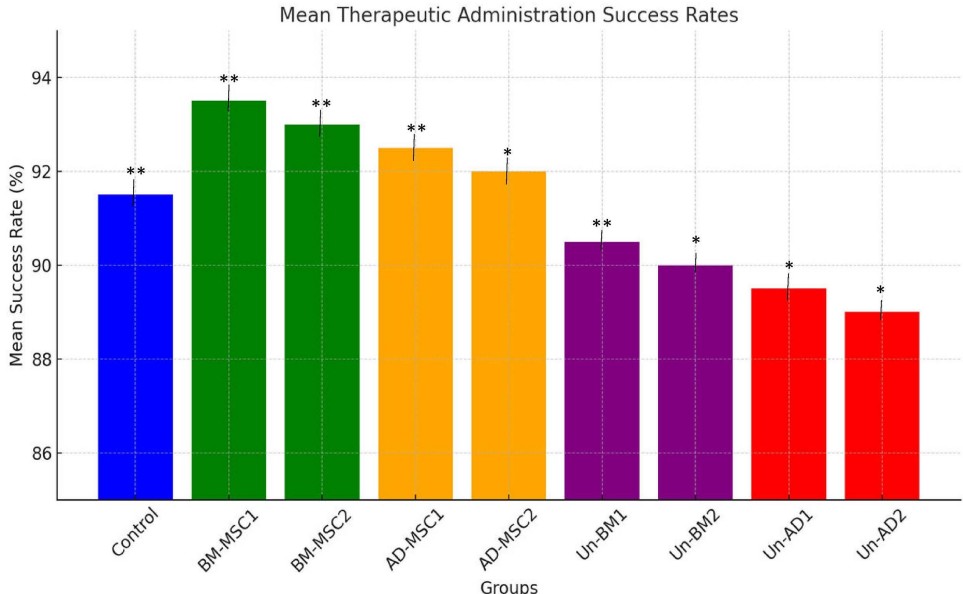

**Fig 5. Therapeutic Administration Success Rates.** Bar chart representing the Mean Therapeutic Administration Success Rates for the control group, treated groups (BM-MSC1, BM-MSC2, AD-MSC1, AD-MSC2), and untreated groups (Un-BM1, Un-BM2, Un-AD1, Un-AD2). Treated groups show an increasing trend, while untreated groups show a decreasing trend. Mesenchymal stem cells (MSCs); bone marrow-derived mesenchymal stem cells (BM-MSCs) and adipose-derived mesenchymal stem cells (AD-MSCs). Mice used n = 12. One-Way ANOVA with statistical significance at $p < 0.05$ (*$p < 0.05$, **$p < 0.01$, and ***$p < 0.001$).

groups (more than two-fold changes). The anti-inflammatory and pain-relieving effects of BM-MSC1 showed a better response among the treatment groups.

## Outcome assessment of cartilage and bone integrity

Fig 7 displays histological and immunohistochemical analyses from different experimental groups using specific staining methods. These methods help evaluate the quality of cartilage regeneration, bone erosion, and ECM composition in the control group and treatment groups using various stem cell sources.

H&E staining reveals differences in cartilage structure and chondrocyte organization across groups (Fig 7). The control group exhibits regular cartilage morphology with well-organized chondrocytes in lacunae. In BM-MSC1 and BM-MSC2 groups, there is improved cellularity and improved organization of chondrocytes, indicating significant carti-lage repair. The AD-MSC1 and AD-MSC2 groups show less organization and cellularity compared to BM-MSC groups, but still show improvement relative to the control, suggesting relative regenerative activity. However, untreated groups (Un-BM1, Un-BM2, Un-AD1, and Un-AD2) show less collagen development, which indicates reduced chondrogenic potential.

Masson's trichrome staining shows collagen distribution and ECM remodeling (Fig 7). The control group displays minimal, blue-stained collagen, indication of limited ECM deposition. In contrast, BM-MSC1 and BM-MSC2 groups show intense collagen staining, reflecting robust ECM synthesis and remodeling. AD-MSC1 and AD-MSC2 groups display relative collagen deposition, less prominent than BM-MSC groups, indicating lower but still measurable ECM restoration. Masson's trichrome staining indicates higher collagen content in BM-MSC-treated groups (BM-MSC1 and BM-MSC2), while untreated groups (Un-BM1, Un-BM2, Un-AD1, and Un-AD2) display less collagen manifestation, which suggest reduced chondrogenic potential.

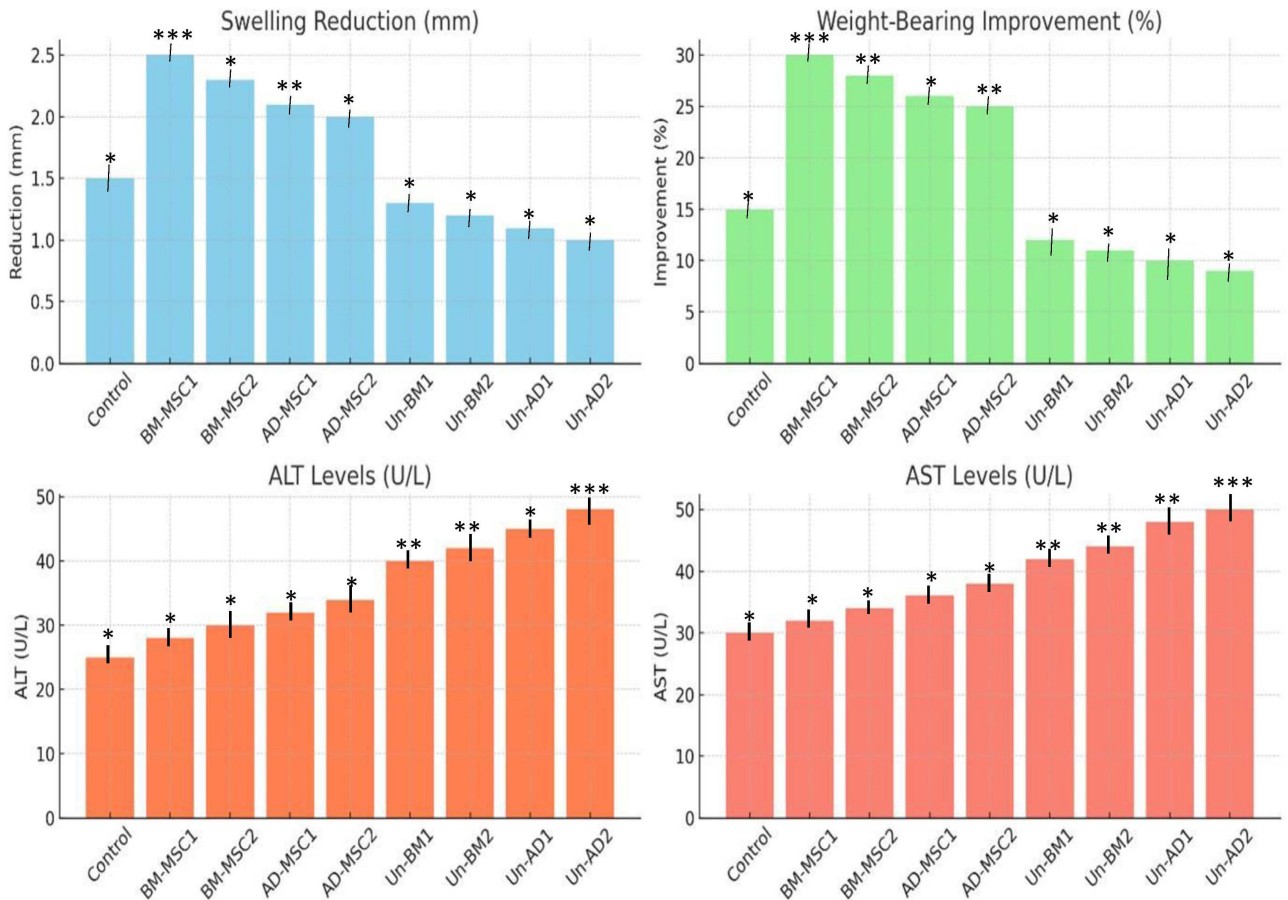

**Fig 6. Result of Assessment of Inflammation and Pain.** Note: BM-MSC1, BM-MSC2, AD-MSC1, AD-MSC2 representing treated groups, while Un-BM1, Un-BM2, Un-AD1, Un-AD2 represent untreated groups. Mesenchymal stem cells (MSCs); bone marrow-derived mesenchymal stem cells (BM-MSCs) and adipose-derived mesenchymal stem cells (AD-MSCs). Mice used was n = 12. One-Way ANOVA with statistical significance at p < 0.05 (*p < 0.05, **p < 0.01, and ***p < 0.001).

**Table 3. Pain and Inflammation expressions.**

| Group | Joint Swelling (mm) | Inflammation Score (0–3) | Pain Sensitivity (g) | IL-6 (pg/mL) | TNF-α (pg/mL) | Gene Expression (Fold Change) |
|---|---|---|---|---|---|---|
| Control | 0.6±0.1 [a] | 0 [a] | 15.2±1.1 [a] | 20±3 [a] | 18±2 [a] | 1.0±0.2 [a] |
| BM-MSC1 | 0.7±0.1 [a] | 1 [ab] | 14.5±1.0 [a] | 24±4 [a] | 19±3 [a] | 1.0±0.3 [a] |
| BM-MSC2 | 0.8±0.2 [a] | 1 [ab] | 13.0±0.9 [a] | 32±5 [a] | 20±4 [a] | 1.3±0.2 [a] |
| AD-MSC1 | 1.1±0.1 [b] | 1 [ab] | 11.5±1.1 [b] | 47±4 [a] | 32±3 [b] | 2.0±0.2 [b] |
| AD-MSC2 | 1.1±0.2 [b] | 1 [ab] | 11.9±1.0 [b] | 42±5 [b] | 36±4 [b] | 2.1±0.3 [b] |
| Un-BM1 | 1.3±0.2 [bc] | 2 [c] | 8.5±0.9 [C] | 75±6 [bc] | 65±5 [C] | 2.5±0.3 [C] |
| Un-BM2 | 1.4±0.2 [c] | 2 [c] | 8.0±1.0 [C] | 80±7 [c] | 70±6 [C] | 2.8±0.4 [C] |
| Un-AD1 | 1.35±0.1 [c] | 2 [c] | 8.2±1.1 [C] | 78±6 [bc] | 68±5 [C] | 2.6±0.3 [C] |
| Un-AD2 | 1.4±0.2 [c] | 3 [d] | 7.8±1.2 [C] | 85±8 [c] | 75±7 [d] | 3.0±0.5 [d] |

Note: Mice used n = 12. One-Way ANOVA with statistical significance at p < 0.05 (*p < 0.05, **p < 0.01, and ***p < 0.001). [a-d]Tukey's post hoc test (for multiple comparisons) with different letters in the same column significant differed at p < 0.05.

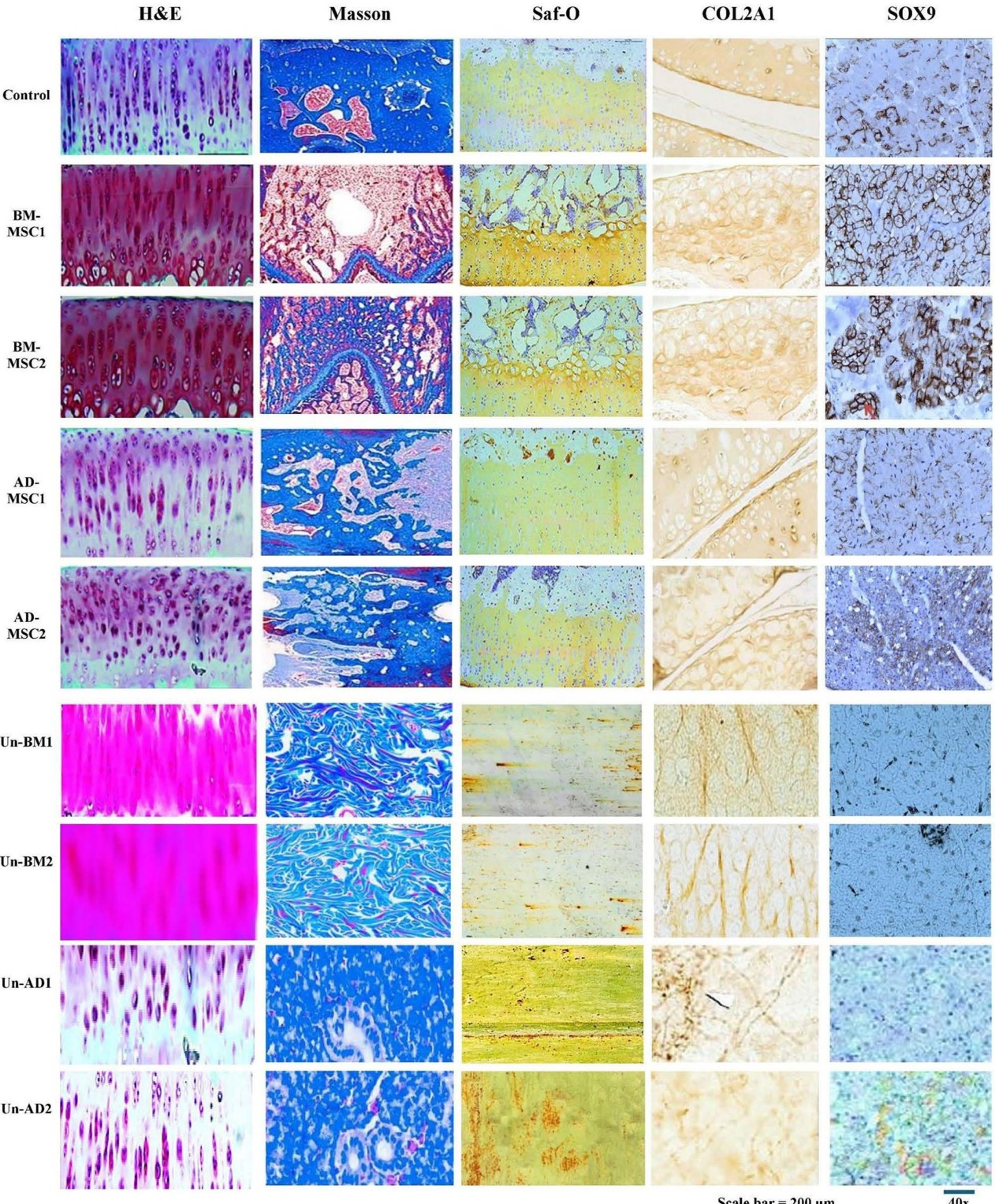

**Fig 7. H&E, Masson's trichrome, Saf-O, COL2A1, and SOX9 of cartilage regeneration, bone erosion, and extracellular matrix (ECM) composition in the control and treatment groups using diverse stem cell sources.** Scale bar = 200 μm (FOV ≈ 0.5 mm) and magnification = 40x. Mice used n = 12. Sample used = femurs and tibiae.

Saf-O staining shows proteoglycan content, a key component of cartilage ECM (Fig 7). The control group shows light staining, signifying limited proteoglycan presence. BM-MSC1 and BM-MSC2 groups demonstrate strong red/orange staining, indicating significant proteoglycan synthesis and cartilage matrix repair. AD-MSC1 and AD-MSC2 groups display weaker staining, but still stronger than the control, suggesting fair ECM restoration. However, untreated groups show weak staining, which indicates poor cartilage development, as compared to BM-MSC groups.

Type II collagen, a hallmark of cartilage ECM, is weakly expressed in the control group, as shown by faint brown staining. In BM-MSC1 and BM-MSC2 groups, there is robust COL2A1 staining, indicating high levels of type II collagen deposition essential for cartilage repair. AD-MSC1 and AD-MSC2 groups show moderate expression of COL2A1, signifying lower but still notable cartilage-specific ECM synthesis compared to BM-MSC groups. The untreated groups show little to no COL2A1 expression. The control group reveals poor expression of SOX9, which indicates chondrogenic differentiation and lack of substantial repair activity. BM-MSC1 and BM-MSC2 groups display strong nuclear staining of SOX9, indicating active chondrogenesis and robust repair potential. In contrast, the untreated groups display weak or absent staining, in line with their poor cartilage-forming potential. In comparison, AD-MSC1 and AD-MSC2 groups display relative SOX9 expression, suggesting a less potent chondrogenic response compared to BM-MSC groups but still higher than the control.

## Outcome assessment of immune modulation

Table 4 presents outcome assessment of immune modulation. BM-MSC1 achieves the highest cytokine modulation (40% change), coupled with a significant reduction in pro-inflammatory cytokines (55%) with an impressive increase in anti-inflammatory cytokines (60%). These changes are consistent with its superior Treg/macrophage ratio and improved NK cell activation, showing its ability to restore immune balance and reduce inflammation effectively. BM-MSC2 follows closely, while AD-MSC1 and AD-MSC2 also show significant, though less pronounced and improvements. In comparison, the untreated groups display minimal to negligible changes, with cytokine modulation barely reaching 10%, pro-inflammatory cytokine reduction under 8%, and anti-inflammatory cytokine increases is below 5%. Their Treg/macrophage

**Table 4. Outcome assessment of immune modulation.**

| Group | Cytokine Modulation (% Change) | Treg/Macrophage Ratio | Pro-Inflammatory Cytokines (% Reduction) | Anti-Inflammatory Cytokines (% Increase) | CD4+/CD8+Ratio | NK Cell Activation (% Change) |
|---|---|---|---|---|---|---|
| Control | 15 [c] | 1.5 | 10 [de] | 12 [c] | 1.8 [b] | 5 [de] |
| BM-MSC1 | 40 [a] | 3.2 [a] | 55 [a] | 60 [a] | 2.5 [a] | 25 [a] |
| BM-MSC2 | 38 [a] | 3.0 [a] | 50 [a] | 55 [a] | 2.4 [a] | 23 [a] |
| AD-MSC1 | 35 [b] | 2.8 [ab] | 45 [a] | 50 [b] | 2.3 [a] | 20 [ab] |
| AD-MSC2 | 33 [b] | 2.7 [ab] | 42 [ab] | 48 [b] | 2.2 [a] | 18 [ab] |
| Un-BM1 | 10 [d] | 1.2 [c] | 8 [de] | 5 [e] | 1.6 [b] | 3 [f] |
| Un-BM2 | 8 [de] | 1.1 [c] | 6 [e] | 4 [e] | 1.5 [b] | 2 [f] |
| Un-AD1 | 6 [e] | 1.0 [c] | 5 [e] | 3 [e] | 1.4 [b] | 1 [f] |
| Un-AD2 | 5 [ef] | 0.9 [e] | 4 [ef] | 2 [f] | 1.3 [b] | 0.5 [g] |

Note: Pro-Inflammatory Cytokines (% Reduction): Indicates the reduction in inflammatory cytokines (e.g., TNF-α, IL-6). Treated groups show significant reductions, with BM-MSC1 being the most effective. Anti-Inflammatory Cytokines (% Increase): Reflects the increase in cytokines like IL-10. Treated groups show significant improvements, with BM-MSC1 leading the effects. CD4+/CD8+Ratio: Represents T-cell balance. Treated groups have improved ratios, indicating a more regulated immune response. NK Cell Activation (% Change): Shows changes in natural killer (NK) cell activation. Treated groups exhibit higher activation, with BM-MSC1 showing the greatest increase. BM-MSC1, BM-MSC2, AD-MSC1, AD-MSC2 representing treated groups, while Un-BM1, Un-BM2, Un-AD1, Un-AD2 represent untreated groups. Mesenchymal stem cells (MSCs); bone marrow-derived mesenchymal stem cells (BM-MSCs) and adipose-derived mesenchymal stem cells (AD-MSCs). Mice used n=12. One-Way ANOVA with statistical significance at $p < 0.05$. [a-g]Tukey's post hoc test (for multiple comparisons) with different letters in the same column significant differed at $p < 0.05$.

ratios and CD4 + /CD8 + balance remain suboptimal, further reflecting their inability to regulate immune activity effectively. This clear difference shows the therapeutic advantage of MSC-based treatments, particularly BM-MSCs, in modulating the immune system, reducing inflammation, and promoting a more positive immune environment. These results strongly suggest that BM-MSC1 has significant promise in addressing immune dysregulation and inflammation-related conditions.

## Safety evaluation

Table 5 displays the results of safety evaluation. The results display a clear distinction in safety profiles between treated and untreated groups, indicating the low toxicity and high tolerability of MSC-based therapies. BM-MSC1 and BM-MSC2 treated groups maintained near-normal levels across all biomarkers, such as ALT, AST, creatinine, and urea, indicating minimal stress on liver and kidney function. BM-MSC1 with the lowest ALT (28 U/L) and AST (32 U/L) levels, along a near-perfect histology score (0.5), stands out as the safest treatment option. AD-MSC1 and AD-MSC2 also show positive safety profiles, though their biomarker levels and histology scores are slightly higher compared to BM-MSCs, suggesting a marginally impact on organ function. In contrast, untreated groups displayed significant elevations in ALT, AST, creatinine, and urea, with levels far exceeding those observed in treated and control groups. However, Un-AD2 showed the highest ALT (48 U/L) and AST (50 U/L) levels, together with a severe histology score of 4.0, indicating substantial liver and kidney damage as well as tissue abnormalities. These findings show the progressive organ dysfunction associated with untreated conditions.

## Comparative analysis

The comparative analysis shows clear therapeutic advantages of MSC-based treatments, with BM-MSC1 emerging as the most effective option for reducing inflammation and promoting cartilage repair (Table 6). BM-MSC1 achieved the highest effect size for both inflammation reduction (45%) and cartilage repair (50%), demonstrating its strong potential to address the underlying issues in inflammatory and degenerative conditions. Subsequently, BM-MSC2 is slightly lower, but still

**Table 5. Safety evaluation.**

| Group | ALT (U/L) | AST (U/L) | Creatinine (mg/dL) | Urea (mg/dL) | Histology Score (0–5) [#] |
|---|---|---|---|---|---|
| Control | 25 [d] | 30 [c] | 0.8 [cd] | 15 [d] | 0 |
| BM-MSC1 | 28 [cd] | 32 [c] | 0.9 [c] | 16 [cd] | 0.5 [d] |
| BM-MSC2 | 30 [cd] | 34 [c] | 0.9 [c] | 17 [c] | 0.7 [d] |
| AD-MSC1 | 32 [c] | 36 [b] | 1.0 [c] | 18 [c] | 1.0 [c] |
| AD-MSC2 | 34 [c] | 38 [b] | 1.1 [c] | 19 [c] | 1.2 [c] |
| Un-BM1 | 40 [ab] | 42 [ab] | 1.4 [ab] | 22 [ab] | 2.5 [b] |
| Un-BM2 | 42 [ab] | 44 [ab] | 1.5 [ab] | 23 [a] | 3.0 [b] |
| Un-AD1 | 45 [a] | 48 [a] | 1.6 [a] | 24 [a] | 3.5 [a] |
| Un-AD2 | 48 [a] | 50 [a] | 1.8 [a] | 25 [a] | 4.0 [a] |

**Note:** ALT (Alanine Aminotransferase, U/L): Elevated levels indicate potential liver damage. Treated groups maintain values close to normal (28–34 U/L), while untreated groups show higher elevations (40–48 U/L). AST (Aspartate Aminotransferase, U/L): Indicates liver or muscle damage. Similar to ALT, treated groups exhibit moderate levels (32–38 U/L), while untreated groups show significant increases (42–50 U/L). Creatinine (mg/dL): Reflects kidney function. Treated groups maintain levels close to normal (0.9–1.1 mg/dL), whereas untreated groups exhibit elevated values (1.4–1.8 mg/dL). Urea (mg/dL): Another marker for kidney function. Treated groups show mild increases (16–19 mg/dL), while untreated groups have higher values (22–25 mg/dL). Histology Score (0–5): Represents organ tissue integrity, with 0 indicating no abnormalities and 5 indicating severe damage. Treated groups exhibit minimal changes (0.5–1.2), whereas untreated groups show significant damage (2.5–4.0). BM-MSC1, BM-MSC2, AD-MSC1, AD-MSC2 representing treated groups, while Un-BM1, Un-BM2, Un-AD1, Un-AD2 represent untreated groups. Mesenchymal stem cells (MSCs); bone marrow-derived mesenchymal stem cells (BM-MSCs) and adipose-derived mesenchymal stem cells (AD-MSCs). Mice used n = 12. Two-Way ANOVA with statistical significance at p < 0.05. [a-g]Tukey's post hoc test (for multiple comparisons) with different letters in the same column significant differed at p < 0.05. [#]Bone and Cartilage Repair Scoring (Modified Pritzker Score): 0–5 scale: Normal cartilage (0) to complete cartilage erosion (5).

**Table 6. Comparative analysis.**

| Group | Inflammation Reduction (Effect Size, %) | Cartilage Repair (Effect Size, %) | Safety Differences (ALT, AST) | ANOVA p-value (Efficacy) |
|---|---|---|---|---|
| Control | 10 [f] | 12 [e] | ALT: 25, AST: 30 | 0.001 |
| BM-MSC1 | 45 [a] | 50 [a] | ALT: 28, AST: 32 | <0.001 |
| BM-MSC2 | 42 [b] | 48 [a] | ALT: 30, AST: 34 | <0.001 |
| AD-MSC1 | 38 [c] | 42 [b] | ALT: 32, AST: 36 | 0.002 |
| AD-MSC2 | 35 [d] | 40 [b] | ALT: 34, AST: 38 | 0.004 |
| Un-BM1 | 15 [e] | 18 [d] | ALT: 40, AST: 42 | 0.05 |
| Un-BM2 | 12 [f] | 15 [c] | ALT: 42, AST: 44 | 0.08 [NS] |
| Un-AD1 | 10 [f] | 14 [cd] | ALT: 45, AST: 48 | 0.09 [NS] |
| Un-AD2 | 8 [g] | 12 [e] | ALT: 48, AST: 50 | 0.1 [NS] |

Note: Inflammation Reduction (Effect Size, %): Treated groups show significant reductions in inflammation, with BM-MSC1 (45%) being the most effective, followed by BM-MSC2 and AD-MSC therapies. Untreated groups show minimal improvement. Cartilage Repair (Effect Size, %): BM-MSC1 leads with a 50% effect size in cartilage repair, with BM-MSC2 and AD-MSC therapies also showing substantial improvement. Untreated groups exhibit limited repair (12–18%). Safety Differences (ALT, AST): ALT and AST levels in treated groups remain close to control values, indicating low safety risks. Untreated groups show much higher ALT and AST levels, reflecting potential organ stress or damage. ANOVA p-value (Efficacy): Statistical significance of the efficacy between groups. Treated groups show highly significant p-values (<0.01), particularly for BM-MSC therapies, indicating superior efficacy compared to untreated groups. BM-MSC1, BM-MSC2, AD-MSC1, AD-MSC2 representing treated groups, while Un-BM1, Un-BM2, Un-AD1, Un-AD2 represent untreated groups. Mesenchymal stem cells (MSCs); bone marrow-derived mesenchymal stem cells (BM-MSCs) and adipose-derived mesenchymal stem cells (AD-MSCs). Mice used n = 12. One-Way ANOVA with statistical significance at p < 0.05. (*p < 0.05, **p < 0.01). [a-g]Tukey's post hoc test (for multiple comparisons) with different letters in the same column significant differed at p < 0.05. [NS]Symbolizes "Not Significant'.

significant. AD-MSC also showed significant efficacy, though their results were less pronounced compared to BM-MSCs. Statistical analysis using ANOVA further reinforces these findings, with p-values below 0.01 for the treated groups, confirming that the observed improvements are both substantial and statistically significant. In contrast, untreated groups showed minimal effect sizes for inflammation reduction (8–18%) and cartilage repair (12–18%), emphasizing the limitations of leaving these conditions untreated.

### RNA sequencing

The real-time PCR CT results reveal differential gene expression throughout the experimental groups (Fig 8). The housekeeping gene GAPDH showed constant CT values across all groups, confirming reliable normalization. Lower CT indicate higher gene expression hence higher inflammatory activity. BM-MSC1 and BM-MSC2 treated groups displayed moderate CT, which indicates a controlled inflammatory response. Conversely, the untreated groups (Un-BM1, Un-BM2, Un-AD1, Un-AD2) have the lowest CT values, indicating increased inflammation and inflammatory gene expression. In comparison to untreated groups, treated groups (BM-MSC1, BM-MSC2, AD-MSC1, and AD-MSC2) showed higher CT values for IL-6, TNF-α, and pro-inflammatory genes, showing lower expression of these inflammatory markers and signifying the effective immunomodulatory effects of the MSC intervention.

The RNA sequencing analysis reveals a profound effect of MSC-based therapies on key gene expression profiles, showing their therapeutic potential (Table 7). BM-MSC1 leads in efficacy, with the highest upregulation of anti-inflammatory (IL-10), cartilage repair (COL2A1, SOX9), and angiogenesis (VEGF) markers. This group also effectively suppressed pro-inflammatory (TNF-α) and cartilage degradation (MMP-13) genes, revealing its robust ability to mitigate inflammation and promote tissue repair. BM-MSC2 showed similar trends, albeit with lower fold-changes across key markers. AD-MSC1 and AD-MSC2 also showed significant upregulation of therapeutic genes, though their effects were less distinct compared to BM-MSCs, indicating lower capacity for immune modulation and tissue regeneration. Untreated groups, on the other hand, exhibited heightened expression of TNF-α and MMP-13, indicating persistent inflammation and

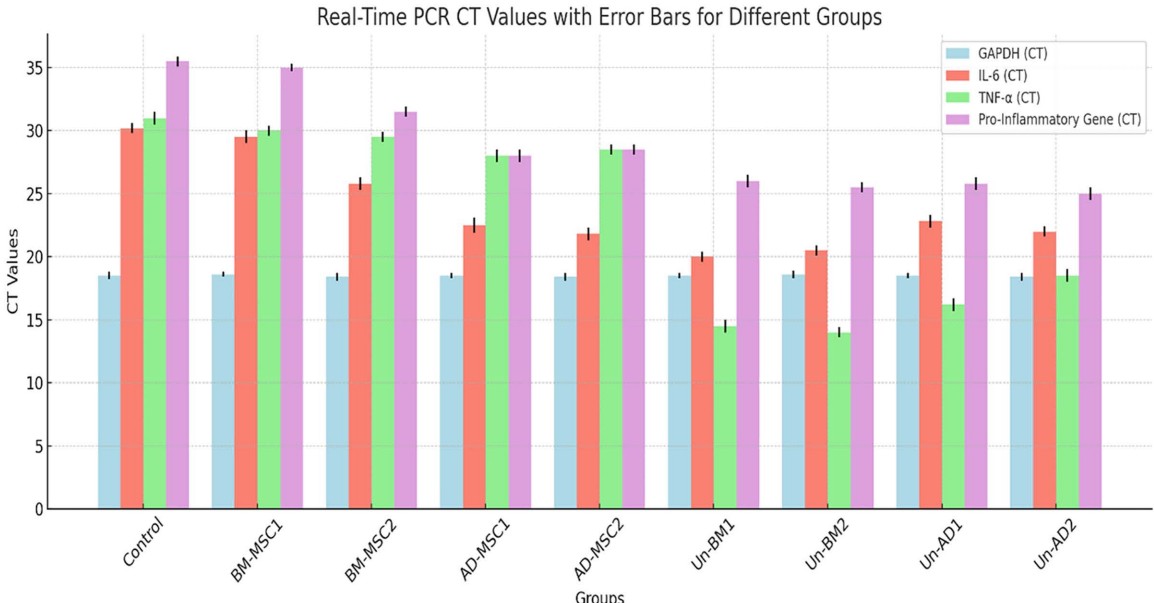

**Fig 8. Real-time PCR Cycle Threshold (CT).** BM-MSC1, BM-MSC2, AD-MSC1, AD-MSC2 representing treated groups, while Un-BM1, Un-BM2, Un-AD1, Un-AD2 represent untreated groups. Mice used n = 12.

**Table 7. Key gene expression profiles.**

| Gene | Gene Function | Expression Before Administration | Expression After Administration | Fold Change (After/Before) | Biological Implication |
|---|---|---|---|---|---|
| IL6 | Pro-inflammatory cytokine | Low (Baseline) | High | +5.2 | Indicates activation of inflammatory pathways. |
| TNF-α | Pro-inflammatory cytokine | Moderate | High | +3.8 | Suggests involvement in immune response post-administration |
| COL1A1 | Collagen synthesis (extra-cellular matrix) | High | Moderate | −1.6 | Reflects potential aging or tissue remodeling effects. |
| MMP9 | Matrix metalloproteinase (ECM degradation) | Low | Moderate | +2.7 | Suggests increased tissue remodeling and repair activity. |
| CDKN2A (p16) | Senescence marker | Moderate | High | +4.5 | Indicates cellular aging and senescence in the MSCs. |
| SIRT1 | Anti-aging regulator | High | Low | −2.3 | Downregulation may reflect aging-associated gene expression. |
| TGFB1 | Anti-inflammatory and ECM repair | Moderate | High | +2.8 | Suggests activation of tissue repair mechanisms. |
| CXCL8 (IL8) | Chemokine (inflammatory signaling) | Low | High | +6.1 | Indicates increased recruitment of immune cells to the site. |

Note: IL6 (Interleukin-6), TNF-α (Tumor Necrosis Factor-alpha), COL1A1 (Collagen Type I Alpha 1 Chain), MMP9 (Matrix Metalloproteinase-9), CDKN2A (p16, Cyclin-Dependent Kinase Inhibitor 2A), SIRT1 (Sirtuin 1), TGFB1 (Transforming Growth Factor Beta 1), and CXCL8 (IL8, C-X-C Motif Chemokine Ligand 8, Interleukin-8).

cartilage degradation. Downregulation of IL-10, SOX9, and VEGF further confirms lack of regenerative activity and worsening tissue damage in these groups.

Fig 9 displaying the fold-change in gene expression levels for various markers (IL-10, COL2A1, TNF-α, SOX9, MMP-13, VEGF, and TGF-β1) across different groups (Control, BM-MSC1/2, AD-MSC1/2, and untreated groups). It reveals the elevated expression of anti-inflammatory and cartilage-repair markers (IL-10, COL2A1, SOX9, TGF-β1) in MSC-treated groups compared to controls and untreated groups. However, pro-inflammatory (TNF-α) and matrix-degrading markers (MMP-13) are significantly higher in untreated groups, indicating the therapeutic potential of MSCs in modulating inflammation and promoting tissue repair.

Western Blot analysis was used to validate non-specific binding observed in BM-MSC groups for Sox9 expression in order to complement the PCR data (Fig 10). The BM-MSC1 group had the highest SOX9 protein expression, followed by the control group. In contrast, BM-MSC2 and AD-MSC1 had lower expression levels. However, GAPDH expression was constant across all groups validates the expression. The measurement of collagen IV levels reveals a notable rise in the BM-MSC1 group, around three folds more than the control, suggesting increased production of matrix proteins. Conversely, the amounts of collagen IV were significantly reduced in BM-MSC2 and AD-MSC1. According to these results, BM-MSC1 may be more effective in increasing matrix protein regulation, which could indicate better differentiation potential or therapeutic efficacy.

## Engraftment imaging

Fig 11 presents the results of the engraftment of MSCs in a mice model over time, measured through bioluminescent imaging. Over time, mice treated with BM-MSC1 and BM-MSC2 show higher bioluminescent signals compared to the other groups, suggesting better engraftment and persistence of these cells in vivo (Fig 7a). AD-MSC-treated mice show relative signal, while the control group lacks significant bioluminescence, as expected. Fig 7b revealed that BM-MSC1 achieved the highest mean flux, followed by BM-MSC2. The AD-MSC groups exhibit lower flux values, though still above the control. Fig 7c displays specific organ distribution of MSCs at weeks 4 and 5, with signals predominantly in the lungs, liver, and gastrosplenic ligament. These results collectively indicate that BM-MSCs engraft more robustly than AD-MSCs, and organ-specific localization varies slightly between groups. This suggests that BM-MSCs might have superior therapeutic potential due to their improved persistence and distribution in target tissues.

The results also reveal significant differences in engraftment, retention, and interaction quality of MSCs across the different groups (Table 8). BM-MSC1 achieved the highest scores, with excellent engraftment (8/10), strong host tissue interaction (9/10), and the highest retention rate (85%). These findings suggest that BM-MSC1 cells integrate seamlessly into host tissue and remain highly functional, making them the most promising group for therapeutic applications. BM-MSC2 has strong outcomes, demonstrating its reliable, while has less effective (Engraftment: 7/10, Host Interaction: 8/10, Retention: 80%). AD-MSCs show moderate efficacy compared to BM-MSCs. AD-MSC1 achieved engraftment score of 6/10, host interaction score of 8/10, and a retention rate of 75%, indicating a good ability to interact with host tissue but a slightly reduced persistence. Untreated MSC groups (Un-BM and Un-AD) performed poorly in all metrics, with low engraftment scores (1–3/10), weak host interaction (2–4/10), and retention rates between 25% and 45%. Overall, BM-MSCs display the highest therapeutic potential, particularly BM-MSC1, while untreated MSCs highlight the importance of intervention strategies to maximize cell performance.

## Discussion

This study provides a thorough comparative analysis of BM-MSCs and AD-MSCs in treating septic arthritis in aged murine models. The findings show significant differences in therapeutic potential, safety profiles, and underlying molecular mechanisms, offering valuable insights for advancing regenerative therapies, particularly in age-related inflammatory conditions. BM-MSCs, especially BM-MSC1, emerged as the most effective treatment option, showing their exceptional ability

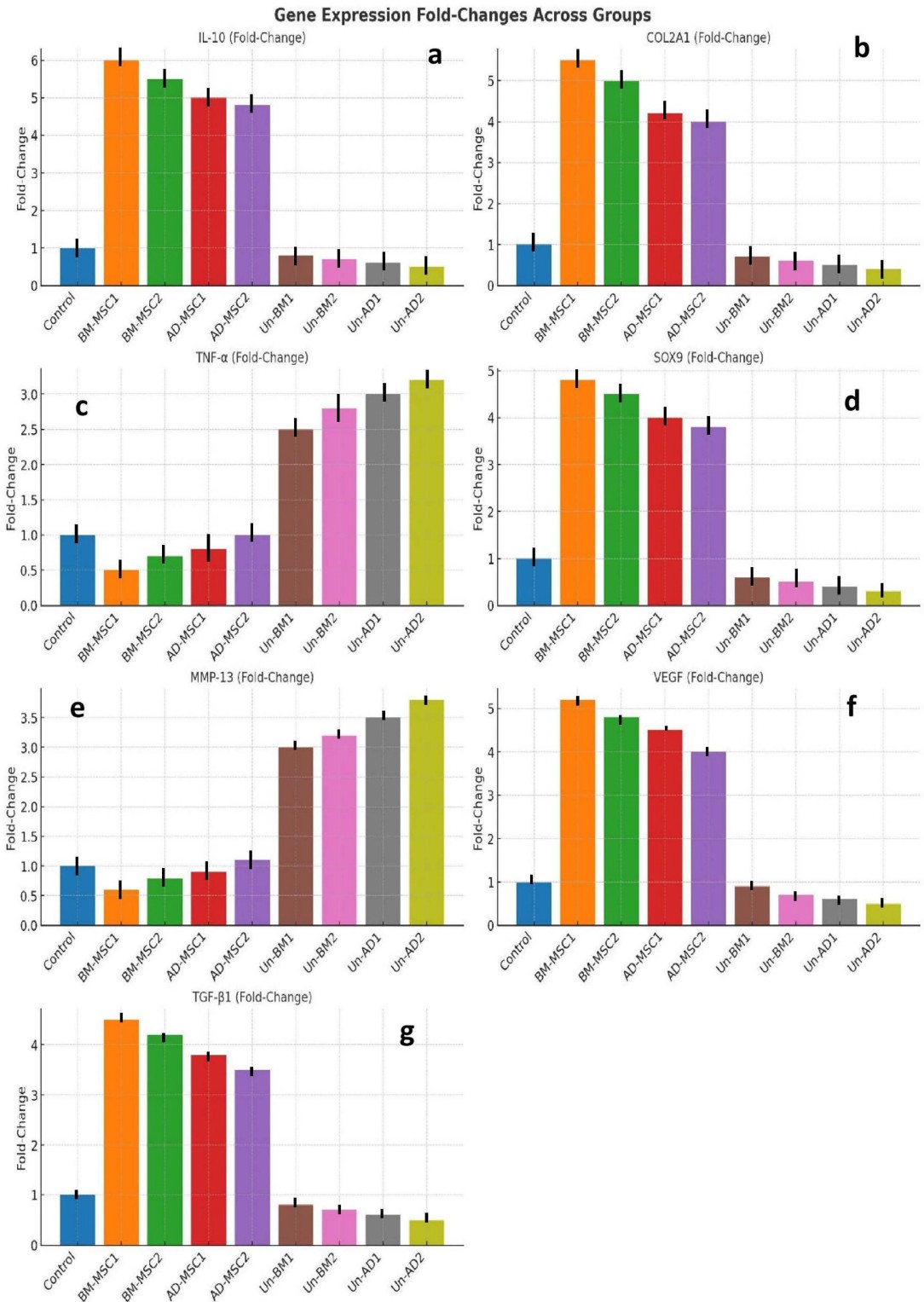

**Fig 9. Fold-change in gene expression levels for various markers: (a) IL-10, (b) COL2A1, (c) TNF-α, (d) SOX9, (e) MMP-13, (f) VEGF, and (g) TGF-β1.** Note: IL-10 (Interleukin-10), COL2A1 (Collagen Type II Alpha 1 Chain), TNF-α (Tumor Necrosis Factor-alpha), SOX9 (SRY-Box Transcription Factor 9), MMP-13 (Matrix Metalloproteinase-13), VEGF (Vascular Endothelial Growth Factor), and TGF-β1 (Transforming Growth Factor Beta 1). Mice

used n = 9. Multiple comparisons were controlled for false discovery rate (FDR) correction (Benjamini-Hochberg). Criterion = adjusted p-value (q) < 0.05 and |log$_2$ fold change| > 1.5 was chosen to identify significantly differentially expressed genes, with 95% CI determined for fold changes.

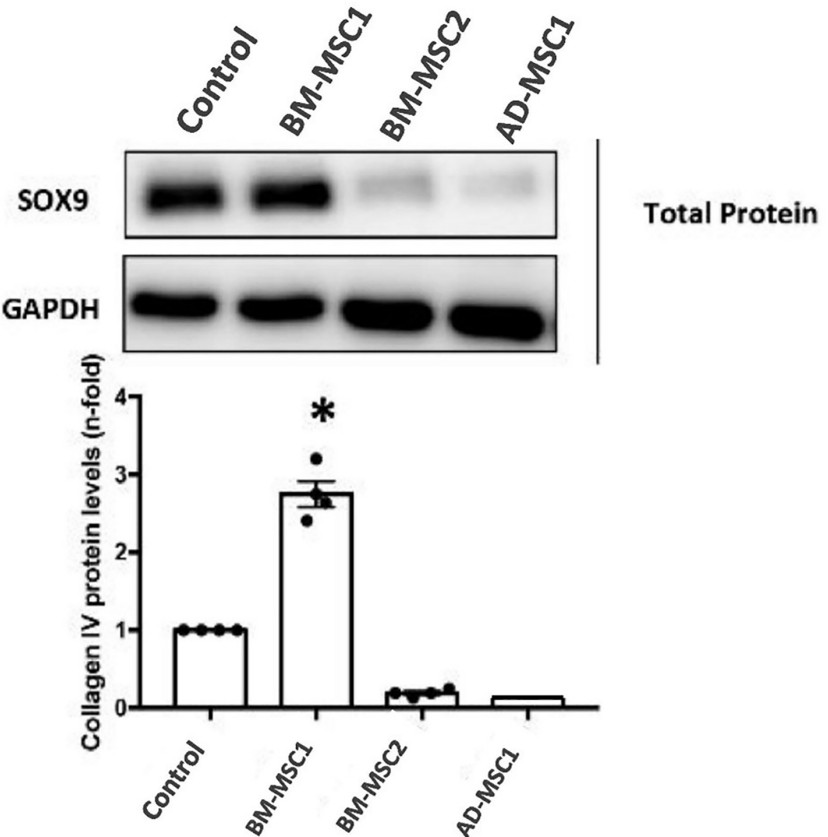

**Fig 10. Western Blot analysis to complement the PCR data confirming SOX9 protein expression in the treated group (BM-MSC).** This further validated the non-specific binding observed in BM-MSC groups for Sox9 expression. Mice used n = 4. One-Way ANOVA with statistical significance at p < 0.05 (*p < 0.05, **p < 0.01, and ***p < 0.001).

to combat inflammation, promote cartilage repair, and modulate immune responses. The study found that BM-MSC1 displayed significant upregulation of key anti-inflammatory and regenerative markers such as IL-10, VEGF, COL2A1, and SOX9, with a concurrent significant downregulation of pro-inflammatory markers such as TNF-α and MMP-13. This unique molecular signature translated into superior clinical outcomes, including 55% reduction in pro-inflammatory cytokines, 60% increase in anti-inflammatory cytokines, and the highest improvements in weight-bearing and joint swelling (30% improvement and 2.5 mm reduction, respectively). BM-MSC2 demonstrated similar trends but with slightly reduced efficacy, reinforcing the robust therapeutic potential of BM-MSCs. This shows superior therapeutic efficacy of BM-MSC1, in treating septic arthritis in aged murine models, demonstrating excellent anti-inflammatory and regenerative capabilities through significant modulation of markers such as IL-10, VEGF, and TNF-α [12].

Comparatively, AD-MSCs offered average therapeutic benefits, showing their potential as an alternative regenerative therapy. AD-MSC1 and AD-MSC2 displayed a significant but lower upregulation of key regenerative markers, such as IL-10 and VEGF, which are important for anti-inflammatory and reparative functions. Despite these benefits, the

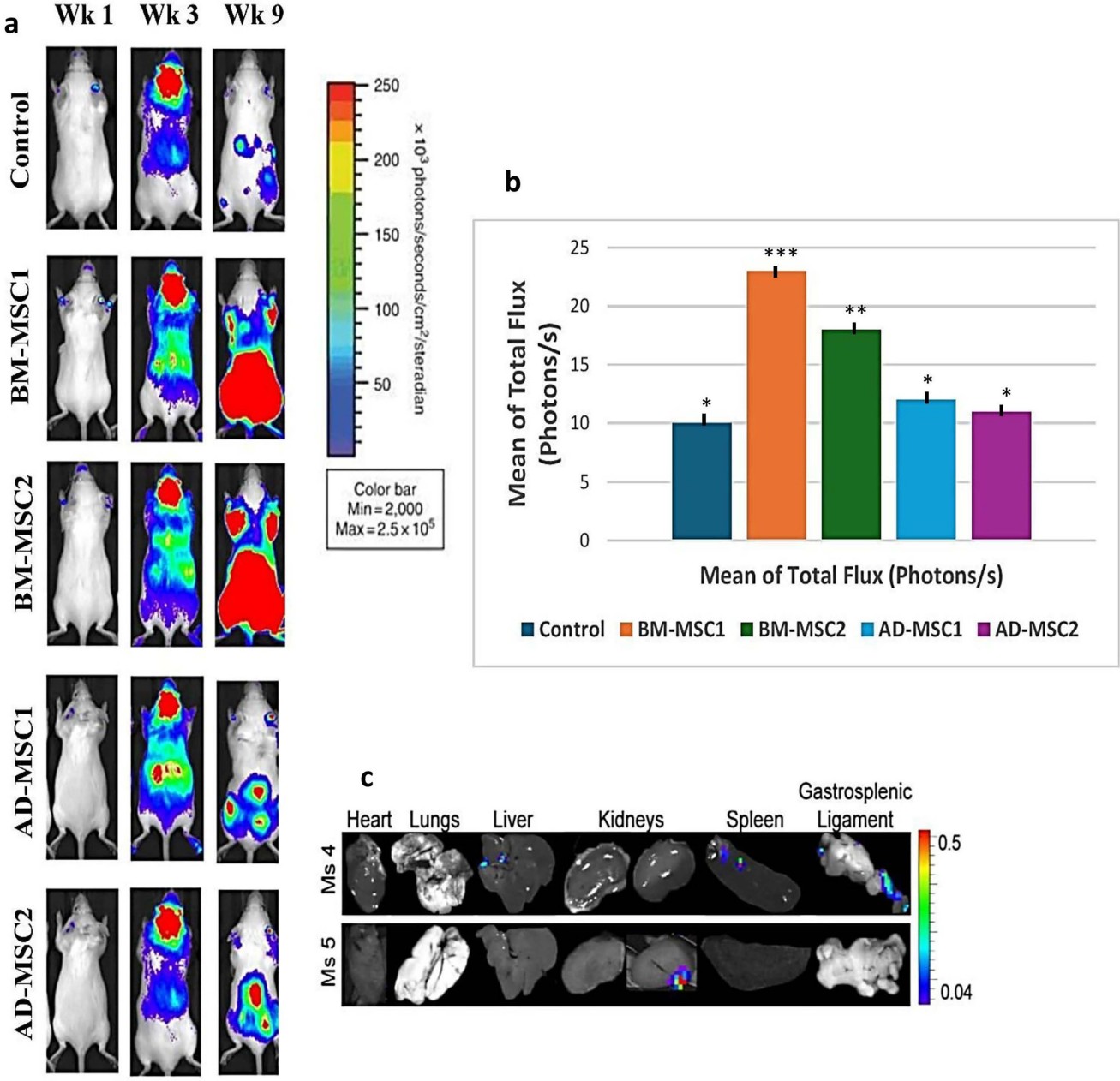

**Fig 11. A longitudinal analysis of MSC engraftment and biodistribution using bioluminescent imaging.** (a) bioluminescence imaging over time, shows bioluminescent signals in mice treated with different MSC types (BM-MSC1, BM-MSC2, AD-MSC1, AD-MSC2) compared to a control group at weeks 1, 3, and 9. The color scale on the images represents the intensity of bioluminescence, indicating cell presence and activity. (b) mean total flux quantification, and (c) organ specific MSC distribution. Mice used n = 12. One-Way ANOVA with statistical significance at $p < 0.05$ (*$p < 0.05$, **$p < 0.01$, and ***$p < 0.001$).

improvements observed with AD-MSCs were less obvious, with swelling reduction limited to 2.1 mm and weight-bearing improvements climaxing at 26%. These results suggest that while AD-MSCs can effectively mitigate inflammation and promote tissue repair, their efficacy may be influenced by donor tissue characteristics and environmental factors in aged

**Table 8. Results of engraftment imaging.**

| Group | Engraftment Level (Score 1–10) | Host Tissue Interaction (Score 1–10) | Retention Rate (% MSCs Detected) | Interaction Quality (Score 1–10) |
|---|---|---|---|---|
| Control | N/A | N/A | 0.0 | N/A |
| BM-MSC1 | 8 [a] | 9 [a] | 85 [a] | 9 [a] |
| BM-MSC2 | 7 [a] | 8 [a] | 80 [a] | 8 [a] |
| AD-MSC1 | 6 [b] | 8 [a] | 75 [a] | 8 [a] |
| AD-MSC2 | 6 [b] | 7 [a] | 72 [b] | 7 [a] |
| Un-BM1 | 3 [c] | 4 [b] | 45 [c] | 5 [b] |
| Un-BM2 | 2 [cd] | 3 [c] | 40 [cd] | 4 [b] |
| Un-AD1 | 1 [d] | 2 [c] | 30 [d] | 3 [c] |
| Un-AD2 | 1 [d] | 2 [c] | 25 [e] | 3 [c] |

Note: Engraftment Level (Score 1–10): Reflects the ability of MSCs to integrate into the host tissue, with higher scores indicating better engraftment. Host Tissue Interaction (Score 1–10): Measures how effectively MSCs interact with the host environment (e.g., immune response modulation, cellular communication). Retention Rate (% MSCs Detected): The percentage of MSCs remaining in the host tissue after a specific period, as detected by imaging techniques. Interaction Quality (Score 1–10): A composite score evaluating the overall quality of MSC-host interactions, considering both engraftment and functional integration. BM-MSC1, BM-MSC2, AD-MSC1, AD-MSC2 representing treated groups, while Un-BM1, Un-BM2, Un-AD1, Un-AD2 represent untreated groups. Mesenchymal stem cells (MSCs); bone marrow-derived mesenchymal stem cells (BM-MSCs) and adipose-derived mesenchymal stem cells (AD-MSCs). Mice used n = 9. One-Way ANOVA with statistical significance at $p < 0.05$. [a-e]Tukey's post hoc test (for multiple comparisons) with different letters in the same column significant differed at $p < 0.05$.

hosts [2,23]. In untreated groups, the severe inflammation, extensive cartilage degradation, and minimal regenerative activity, indicated by elevated TNF-α and MMP-13 levels, suggest the vital need for early MSC intervention to prevent irreversible joint damage [31]. Collectively, these findings confirm that both BM-MSCs and AD-MSCs are reliable and have efficacy for managing septic arthritis, but the superior anti-inflammatory and regenerative capabilities of BM-MSCs make them particularly well-suitable for addressing the complex pathophysiology of the condition in aged individuals.

Ensuring the safety of any therapeutic approach is essential, especially when addressing conditions in vulnerable populations such as the elderly. This study shows the remarkable safety profile of MSC-based treatments, with BM-MSCs performing highly as the most reliable option. Across all treated groups, key indicators of liver and kidney function, including ALT, AST, creatinine, and urea levels, remained within or near normal ranges, while histological analysis revealed minimal abnormalities in major organs. BM-MSC1 emerged as the safest candidate, displaying the lowest ALT (28 U/L) and AST (32 U/L) levels, indication of minimal liver stress, and achieving an impressive histology score of 0.5, reflecting almost no detectable organ damage or systemic toxicity. This is particularly significant in aged models, where underlying fragility often amplifies the risk of adverse effects from treatments. While AD-MSCs also displayed acceptable safety profile, their slightly elevated biomarker levels compared to BM-MSCs suggest a marginally higher systemic burden. Nonetheless, both BM-MSCs and AD-MSCs present a compelling case for safe and effective interventions in inflammatory and degenerative diseases. The superior safety outcomes of BM-MSCs, especially BM-MSC1, show their suitability for regenerative therapies in aging populations, providing confidence that therapeutic benefits can be achieved without negatively affecting patient safety [32,33].

Imaging studies provided visual and quantitative evidence to support these findings. BM-MSC1 achieved the most significant outcomes for engraftment (8/10), host tissue interaction (9/10), and retention rates (85%), signifying its superior capacity to integrate into the damaged joint environment and persist to drive profound therapeutic effects. Studies have suggested that effective integration likely plays a pivotal role in modulating the local immune response [34,35], which is evident by the findings of this study including reducing joint swelling and promoting structural repair. This synergy of molecular and imaging data provides a robust foundation for advancing BM-MSC1 in clinical applications. Moreover, AD-MSCs show functional efficacy but fall short of BM-MSCs in imaging and molecular analyses. Retention rates

for AD-MSC1 and AD-MSC2 are lower than BM-MSCs, indicating less robust ability to persist in damaged joints. Their engraftment and tissue interaction scores are moderate, suggesting limited capacity for sustained engagement with host tissues. Untreated groups show little to no regenerative activity, with persistent inflammation and elevated markers of cartilage degradation. Imaging studies reveal poor integration, with retention rates below 45%. This highlights the need for optimized MSC-based interventions to mitigate severe outcomes associated with untreated conditions.

The findings of this study have significant implications for regenerative medicine, particularly in managing inflammatory and degenerative joint diseases such as septic arthritis in aging populations. BM-MSC1 has displayed a unique balance of efficacy and safety, making them a prime candidate for therapeutic use. Their ability to effectively reduce inflammation, promote cartilage repair, and maintain systemic safety proves their potential as a gold-standard treatment option. AD-MSCs, while effective, may serve as an alternative where bone marrow extraction is not feasible. However, the slightly lower efficacy and safety outcomes suggest that further optimization is needed for AD-MSCs to match the performance of BM-MSCs.

The study demonstrated that BM-MSC performs better than AD-MSC, particularly BM-MSC1, outperform AD-MSCs in terms of efficacy, safety, and residence time in treating septic arthritis in aged murine models. BM-MSC1 showed superior therapeutic efficacy due to its upregulation of anti-inflammatory and regenerative markers, while downregulating pro-inflammatory ones. This result in better outcomes, including 55% reduction in pro-inflammatory cytokines, 60% increase in anti-inflammatory cytokines, and improvements in weight-bearing and joint swelling. Even though, AD-MSCs showed therapeutic benefits but lower in overall efficacy, with less significant improvements in swelling and weight-bearing. BM-MSCs are better suited for achieving significant regenerative outcomes. BM-MSC1 has a superior safety profile due to with minimal liver stress and no detectable organ damage, as it had the lowest ALT and AST levels, indicating minimal liver stress. AD-MSCs had relatively elevated biomarker levels, suggesting a marginally higher systemic burden. Imaging studies showed that BM-MSC1 has superior engraftment, tissue interaction, and retention rates, indicating its ability to integrate into damaged joint environments and persist for sustained therapeutic effects. This integration likely modulates the local immune response, contributing to better structural repair and inflammation control. AD-MSCs have lower retention rates and moderate tissue interaction, making BM-MSCs valuable for long-term regenerative outcomes. BM-MSCs have enhanced differentiation capacity and increased production of bioactive molecules are major factors in their improved efficacy over AD-MSCs. BM-MSCs showed higher multipotency which can be attributed to differentiating into osteogenic (bone-forming) and chondrogenic (cartilage-forming) lineages. They also release additional growth hormones, including as transforming TGF-β, VEGF, and HGF, which are essential for promoting tissue repair and reducing inflammation. Their high immunomodulatory capability, which also enables better control of immune responses, makes them more useful in treating aged related septic arthritis.

## Conclusion

The study reveals the potential of MSCs in treating septic arthritis, especially in elderly populations. BM-MSCs were found to be the most effective in reducing inflammation, repairing cartilage, and modulating immune responses. BM-MSC1 demonstrated significant improvements in joint health and robust molecular adaptations, making it a leading candidate for advanced regenerative therapies. Even though AD-MSCs showed some therapeutic benefits, their outcomes were less evident than BM-MSCs, suggesting they could be an alternative when bone marrow extraction is not feasible. The study also highlighted the strong safety profile of BM-MSCs, displaying minimal toxicity and systemic stress. The study shows the urgent need for effective therapies that directly address the underlying causes of conditions such as septic arthritis.

The study suggests that future research should focus on optimizing the therapeutic potential of MSCs. Even though BM-MSCs have shown superior efficacy, further refinement of protocols is needed to enhance their regenerative and immunomodulatory capacities. Pre-treatment strategies, such as priming MSCs with cytokines, growth factors, or hypoxic conditions, could amplify their therapeutic impact. AD-MSCs showed potential, but targeted optimization through gene

editing, tailored culture techniques, or combination therapies could bridge the gap. Expanding research into larger animal models will provide deeper insights into optimal dosing, biodistribution, and long-term safety. Clinical trials can prioritize elderly patients with septic arthritis, assessing long-term outcomes and addressing the combined effects of aging and inflammation in vulnerable individuals. The study suggests that understanding the mechanisms of MSC action is important for developing effective therapeutic approaches. It suggests that using advanced molecular tools such as single-cell RNA sequencing and proteomics can reveal how MSCs interact with the immune system and inflammatory environments, especially in aged hosts. However, it also emphasizes the need for long-term safety and ethical considerations, including rigorous monitoring for adverse effects and addressing ethical issues such as donor selection and treatment access.

## Supporting information

**S1 File. Supporting Information files (Data).**
(ZIP)

## Acknowledgments

We acknowledge all the support provided by College of Medicine, Tikrit University, Salahaldin, Iraq, in terms of enabling environment for this research.

## Author contributions

**Conceptualization:** Alani Mohanad Khalid Ahmed.

**Data curation:** Alani Mohanad Khalid Ahmed.

**Formal analysis:** Mujahid Khalaf Ali.

**Investigation:** Basma Kh. Alani.

**Methodology:** Alani Mohanad Khalid Ahmed, Mujahid Khalaf Ali.

**Resources:** Mujahid Khalaf Ali.

**Software:** Mujahid Khalaf Ali.

**Supervision:** Alani Mohanad Khalid Ahmed.

**Validation:** Basma Kh. Alani.

**Visualization:** Basma Kh. Alani.

**Writing – original draft:** Alani Mohanad Khalid Ahmed.

**Writing – review & editing:** Alani Mohanad Khalid Ahmed, Basma Kh. Alani.

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
