## [Decision Letter · Decision Letter 0]

21 Feb 2025

Dear Dr. Khalid Ahmed,

We look forward to receiving your revised manuscript.

Kind regards,

Sharun Khan

Academic Editor

PLOS ONE

Journal Requirements:

3. To comply with PLOS ONE submissions requirements, in your Methods section, please provide additional information regarding the experiments involving animals and ensure you have included details on (1) methods of sacrifice, (2) methods of anesthesia and/or analgesia, and (3) efforts to alleviate suffering.

Reviewers' comments:

Reviewer's Responses to Questions

**Comments to the Author**

1. Is the manuscript technically sound, and do the data support the conclusions?

Reviewer #1: Partly

Reviewer #2: Partly

2. Has the statistical analysis been performed appropriately and rigorously?

Reviewer #1: No

Reviewer #2: No

3. Have the authors made all data underlying the findings in their manuscript fully available?

Reviewer #1: Yes

Reviewer #2: No

4. Is the manuscript presented in an intelligible fashion and written in standard English?

Reviewer #1: Yes

Reviewer #2: No

Reviewer #1: Septic arthritis is a disease characterized by rapid joint damage as well as cartilage degeneration, and there is a growing interest in the use of MSCs for the treatment of septic arthritis due to their accessibility and regenerative potential.This study compared the efficacy, safety, and delivery of BM-MSCs and AD-MSCs in the treatment of septic arthritis. However, there are still several issues that need to be addressed.

1. In this study, the authors set up a total of nine subgroups, including control group, treated group(BM-MSC1,BM-MSC2,AD-MSC1,AD-MSC2), and untreated group(Un-BM1,Un-BM2,Un-AD1,Un-AD2). However, there are some places in the manuscript where the descriptions of the groups are not clear enough, for example, the abstract section may not be accurate in describing the groups. In addition, it seems that the relevant results for the model group are missing from the manuscript.

2. The definitions of treated and untreated are not clear in the manuscripts. Please rephrase for better understanding.

3. Please indicate the generation of MSCs used for the experiment. In addition, please provide a morphologic map of the MSCs and a file in FCS format for flow cytometry.

4. Please describe in more detail the experimental details of the section of “Materials and Methods”. In addition, there are some problems with the logic of the language in this section, please rewrite this section.

5. This study compared the efficacy and safety of BM-MSCs with AD-MSCs. Interestingly, BM-MSC1 appears to be the optimal candidate for the treatment of septic osteoarthritis. However, what does “1”, and “2” mean respectively, as well as please elucidate the potential mechanism that produces this phenomenon.

6. Please provide HE staining and histochemical results of the model group and untreated group(Un-BM1,Un-BM2,Un-AD1,Un-AD2) to better visualize the efficacy of MSCs. Please add a scale bar to the HE staining and histochemistry diagrams or write the magnification in the figure legends.

7. Please add symbols to the bar graphs that can represent statistical differences. In addition, please indicate in the figure legends the sample size of animals used for experimental analyses.

8. The study shown that BM-MSC outperforms AD-MSC in terms of efficacy, safety, and residence time. Please explain these phenomena.

Reviewer #2: The authors aimed to address a significant and relevant biological question and have employed robust analytical techniques; however, there are notable areas that require improvement. The manuscript, in its current form, presents significant methodological and reporting deficiencies that must be addressed. Without clear experimental design details, proper stem cell characterization, statistical clarity, and complete data presentation, the study’s conclusions remain unsubstantiated. I recommend major revisions to improve the manuscript’s scientific validity before it can be reconsidered for publication.

The study has potential relevance but is not presented or documented properly, it’s crucial to address the bare minimum requirements to improve its quality and credibility.

• Clearly state the methodology and ensure that all treatment groups are properly defined.

• Revise for clarity and completeness.

• Ensure that all abbreviations are defined upon first use in the abstract and text (e.g., ADMSC and BMSC).

• Line 54: Verify the reference that claims BM-MSCs as the gold standard.

• Line 68: Add a supporting reference.

• Specify the total number of animals used, including their sex and distribution across groups.

• Clearly mention how many animals were used for stem cell isolation.

• The study lacks characterization of stem cell markers—explain why specific markers for ADMSC and BM-MSC were not used.

• Minimum reporting criteria for stem cell characterization have not been met. Include data to support MSC characterization

• The current description lacks clarity. Introduce a separate heading for Experimental Design.

• Clearly define ADMSC1, ADMSC2, BMMSC1, and BMMSC2, explaining their roles in the study.

• Line 127: Add a reference for the dose of stem cells used.

• Line 164: Specify the statistical methods used, including the software, significance level, and confidence interval.

• Mention the primers used or provide a reference for RNA quantification methods.

• The ISCT criteria for MSC characterization have not been fully met—include additional characterization data to confirm MSC identity and differentiate ADMSC from BM-MSC.

• Clearly mention the histological scoring method used.

• Include necessary references for materials and methods.

• Histopathology Figures:

o Ensure scale bars and magnifications are specified.

o Specify the organ being analyzed in the figure legends.

• Sox9 Expression: The non-specific binding observed in BM-MSC groups needs to be clarified. Provide an alternative figure or additional data for validation.

• Fold Change Calculations:

o Specify the method used for fold change calculations.

o Include details on statistical analysis of fold changes.

• Pain & Inflammation Data:

o The manuscript mentions measurements in Materials & Methods, but results are not provided.

o If included in supplementary material, mention this in the main text and provide relevant details.

• Figures 3, 4, and 7: Mark statistical significance on graphs.

• Using only one pro-inflammatory marker is insufficient to draw reliable conclusions. Consider adding more markers to strengthen the findings.

• Give data of all scores used

• Give data of real time RNA quantification results -CT values

**Do you want your identity to be public for this peer review?** For information about this choice, including consent withdrawal, please see our Privacy Policy

Reviewer #1: No

Reviewer #2: No

---

## [Author Response · Author response to Decision Letter 1]

8 May 2025

Response to Reviewer 1 (Report 1)

Septic arthritis is a disease characterized by rapid joint damage as well as cartilage degeneration, and there is a growing interest in the use of MSCs for the treatment of septic arthritis due to their accessibility and regenerative potential. This study compared the efficacy, safety, and delivery of BM-MSCs and AD-MSCs in the treatment of septic arthritis. However, there are still several issues that need to be addressed:

Comments 1: In this study, the authors set up a total of nine subgroups, including control group, treated group (BM-MSC1,BM-MSC2,AD-MSC1,AD-MSC2), and untreated group(Un-

BM1,Un-BM2,Un-AD1,Un-AD2). However, there are some places in the manuscript where the descriptions of the groups are not clear enough, for example, the abstract section may not be accurate in describing the groups. In addition, it seems that the relevant results for the model group are missing from the manuscript.

Response 1: Thank you for the time and effort putting in reviewing and pointing out these. We agree with this comment. Therefore, we have provided complete description of the groups in the abstract and text (page 4-5). All relevant results have been added, please refer to revised version.

Comments 2: The definitions of treated and untreated are not clear in the manuscripts. Please rephrase for better understanding.

Response 2: We agreed with comment. Thus, we have provided a more detailed explanation in the revised file, page 4-5.

Comments 3: Please indicate the generation of MSCs used for the experiment. In addition, please provide a morphologic map of the MSCs and a file in FCS format for flow cytometry.

Response 3: The generation has been provided as suggested (page 5) and we have also provided morphologic map of the MSCs, please refer to page 14-15. The file in FCS format for flow cytometry has been provided as supplementary file.

Comments 4: Please describe in more detail the experimental details of the section of “Materials and Methods”. In addition, there are some problems with the logic of the language in this section, please rewrite this section.

Response 4: The Materials and Methods section has been revised and more details have been added as suggested. Please refer to page 4-12.

Comments 5: This study compared the efficacy and safety of BM-MSCs with AD-MSCs. Interestingly, BM-MSC1 appears to be the optimal candidate for the treatment of septic osteoarthritis. However, what does “1”, and “2” mean respectively, as well as please elucidate the potential mechanism that produces this phenomenon.

Response 5: The “1”, and “2” attached to BM-MSC refer to type 1 and type 2 MSCs. More detail explanations have been provided in second paragraph on page 5 (revised file).

Comments 6: Please provide HE staining and histochemical results of the model group and untreated group(Un-BM1,Un-BM2,Un-AD1,Un-AD2) to better visualize the efficacy of MSCs. Please add a scale bar to the HE staining and histochemistry diagrams or write the magnification in the figure legends.

Response 6: The HE staining and histochemical results for both model group and untreated group have been provided as suggested and the scale bar has been added.

Comments 7: Please add symbols to the bar graphs that can represent statistical differences. In addition, please indicate in the figure legends the sample size of animals used for experimental analyses.

Response 7: Symbols have been added to bar graphs that represent statistical differences and sample size used for experimental analyses have been included across the manuscript.

Comments 8: The study shown that BM-MSC outperforms AD-MSC in terms of efficacy, safety, and residence time. Please explain these phenomena.

Response 8: An explanation has been provided to support BM-MSC outperforms AD-MSC in terms of efficacy, safety, and residence time. Please refer to page 38-39.

Thank and appreciate time and effort that the reviewer put to improve our work.

Response to Reviewer 2 (Report 2)

The authors aimed to address a significant and relevant biological question and have employed robust analytical techniques; however, there are notable areas that require improvement. The manuscript, in its current form, presents significant methodological and reporting deficiencies that must be addressed. Without clear experimental design details, proper stem cell characterization, statistical clarity, and complete data presentation, the study’s conclusions remain unsubstantiated. I recommend major revisions to improve the manuscript’s scientific validity before it can be reconsidered for publication. The study has potential relevance but is not presented or documented properly, it’s crucial to address the bare minimum requirements to improve its quality and credibility.

Comments 1: Clearly state the methodology and ensure that all treatment groups are properly defined. Revise for clarity and completeness.

Response 1: The methodology has been revised for clarity and all treatment groups have been properly defined. Please refer to the revised file.

Comments 2: Ensure that all abbreviations are defined upon first use in the abstract and text (e.g., ADMSC and BMSC).

Response 2: All abbreviations have been defined as suggested.

Comments 3: Please indicate the generation of MSCs used for the experiment. In addition, please provide a morphologic map of the MSCs and a file in FCS format for flow cytometry.

Response 3: The generation has been provided as suggested (page 6) and we have also provided morphologic map of the MSCs, please refer to pages 14-15. The file in FCS format for flow cytometry has been provided as supplementary file.

Comments 4: Line 54: Verify the reference that claims BM-MSCs as the gold standard.

Response 4: The reference has been provided and verified as suggested. The supporting reference has been added. Please refer to line 57-58 in the revised file.

Comments 5: Line 68: Add a supporting reference.

Response 5: The supporting reference has been added. Please refer to line 72-73 in the revised version.

Comments 6: Specify the total number of animals used, including their sex and distribution across groups. Clearly mention how many animals were used for stem cell isolation.

Response 6: Number of animals used including gender have been specified (page 5 line 107-122 and line 171-174).

Comments 7: The study lacks characterization of stem cell markers—explain why specific markers for ADMSC and BM-MSC were not used.

Response 7: The information on characterization of stem cell markers have been provided. Please refer to page 7 and pages 12-13.

Comments 8: Minimum reporting criteria for stem cell characterization have not been met. Include data to support MSC characterization

Response 8: The data to support MSC characterization has been included as suggested. Please refer to page 7 and pages 12-13.

Comments 9: The current description lacks clarity. Introduce a separate heading for Experimental Design.

Response 9: A separate heading for Experimental Design has been introduced as suggested.

Comments 10: Clearly define ADMSC1, ADMSC2, BMMSC1, and BMMSC2, explaining their roles in the study.

Response 10: ADMSC1, ADMSC2, BMMSC1, and BMMSC2 cells have been clearly defined as suggested.

Comments 11: Line 127: Add a reference for the dose of stem cells used.

Line 164: Specify the statistical methods used, including the software, significance level, and confidence interval.

Response 11: References used have been added. Please refer to page 9. The statistical methods used, including the software, significance level, and confidence interval have been provided on pages 12-13.

Comments 12: Mention the primers used or provide a reference for RNA quantification methods.

Response 12: The primers used are provided on page 11.

Comments 13: The ISCT criteria for MSC characterization have not been fully met—include additional characterization data to confirm MSC identity and differentiate ADMSC from BM-MSC.

Response 13: We have provided more details and results in fulfilment of ISCT criteria for MSC characterization. Please refer to page 7-8.

Comments 14: Clearly mention the histological scoring method used.

Response 14: We have provided the histological scoring method used on page 13.

Comments 15: Include necessary references for materials and methods.

Response 15: All necessary references have been included.

Comments 14: Histopathology Figures:

o Ensure scale bars and magnifications are specified.

o Specify the organ being analyzed in the figure legends.

Response 14: Histopathology figures have been improved as suggested.

Comments 15: Sox9 Expression: The non-specific binding observed in BM-MSC groups needs to be clarified. Provide an alternative figure or additional data for validation.

Response 15: Alternative figure has been provided as suggested. Please refer to pages 32-33.

Comments 16: • Fold Change Calculations:

o Specify the method used for fold change calculations.

o Include details on statistical analysis of fold changes.

Response 16: The method used has been included as requested.

Comments 17: Pain & Inflammation Data:

o The manuscript mentions measurements in Materials & Methods, but results are not provided.

o If included in supplementary material, mention this in the main text and provide relevant details.

Response 17: Please refer to page 18-19 and figure 6.

Comments 18: Figures 3, 4, and 7: Mark statistical significance on graphs.

Response 18: The statistical significances have been marked on graphs as requested.

Comments 19: Using only one pro-inflammatory marker is insufficient to draw reliable conclusions. Consider adding more markers to strengthen the findings.

Response 19: Tumor necrosis factor-alpha (TNF-a) and matrix metalloproteinase-13 (MMP-13) pro-inflammatory markers used in this study.

Comments 20: Give data of all scores used

• Give data of real time RNA quantification results -CT values

Response 20: The real time RNA quantification results -CT values have been provided. Please refer to page 28 and fig 8.

Thank and appreciate time and effort that the reviewer put to improve our work.

---

## [Decision Letter · Decision Letter 1]

17 Jun 2025

Dear Dr. Khalid Ahmed,

Thank you for submitting your manuscript to PLOS ONE. After careful consideration, we feel that it has merit but does not fully meet PLOS ONE’s publication criteria as it currently stands. Therefore, we invite you to submit a revised version of the manuscript that addresses the points raised during the review process.

We look forward to receiving your revised manuscript.

Kind regards,

Sharun Khan

Academic Editor

PLOS ONE

Journal Requirements:

Reviewers' comments:

Reviewer's Responses to Questions

**Comments to the Author**

Reviewer #2: All comments have been addressed

2. Is the manuscript technically sound, and do the data support the conclusions?

Reviewer #2: Yes

3. Has the statistical analysis been performed appropriately and rigorously?

Reviewer #2: Yes

4. Have the authors made all data underlying the findings in their manuscript fully available?

Reviewer #2: Yes

5. Is the manuscript presented in an intelligible fashion and written in standard English?

Reviewer #2: Yes

Reviewer #2: The authors have made significant revisions and incorporated the necessary data, improving the overall quality of the manuscript. The study is now close to being suitable for publication in PLOS ONE, pending a few minor revisions outlined below:

1. (Lines 119–120) There is a minor confusion regarding the differentiation between the treated and untreated groups. If my understanding is correct: Treated = septic arthritis + MSC therapy, Untreated = septic arthritis only (no therapy) and Control = PBS

To improve clarity, consider rephrasing the sentence as:

“The treated groups received MSC therapy (BM-MSCs or AD-MSCs) after arthritis induction, while the untreated groups received no therapy following arthritis induction and thus served as positive controls.”

2. Repetition of procedures related to MSC isolation has been observed section on MSC isolation. Please revise this section to eliminate redundancy and streamline the methodology for clarity and conciseness.

3. Ensure standardization of letter casing. Please ensure proper use of upper and lower case letters within sentences. For instance, terms such as "Positive marker" or "Red of Alizarin red" should follow standard sentence case conventions unless referring to specific proper nouns. Revise the manuscript accordingly to maintain consistency in formatting.

Once these minor revisions are addressed, the manuscript will be suitable for publication.

**Do you want your identity to be public for this peer review?** For information about this choice, including consent withdrawal, please see our Privacy Policy

Reviewer #2: No

---

## [Author Response · Author response to Decision Letter 2]

18 Jun 2025

Response to Reviewer 2

The authors have made significant revisions and incorporated the necessary data, improving the overall quality of the manuscript. The study is now close to being suitable for publication in PLOS ONE, pending a few minor revisions outlined below:.

Comments 1: 1. (Lines 119–120) There is a minor confusion regarding the differentiation between the treated and untreated groups. If my understanding is correct: Treated = septic arthritis + MSC therapy, Untreated = septic arthritis only (no therapy) and Control = PBS

To improve clarity, consider rephrasing the sentence as:

“The treated groups received MSC therapy (BM-MSCs or AD-MSCs) after arthritis induction, while the untreated groups received no therapy following arthritis induction and

thus served as positive controls.”.

Response 1: The statement has been revised accordingly. Please refer to the revised file.

Comments 2: Repetition of procedures related to MSC isolation has been observed section on MSC isolation. Please revise this section to eliminate redundancy and streamline the methodology for clarity and conciseness.

Response 2: The section has been revised as suggested.

Comments 3: Ensure standardization of letter casing. Please ensure proper use of upper and lower case letters within sentences. For instance, terms such as "Positive marker" or "Red of

Alizarin red" should follow standard sentence case conventions unless referring to specific proper nouns. Revise the manuscript accordingly to maintain consistency in formatting.

Response 3: The standardization of letter casing has been revised across the manuscript as suggested

Thank and appreciate time and effort that the reviewer put to improve our work.

---

## [Decision Letter · Decision Letter 2]

18 Sep 2025

Thank you for submitting your manuscript to PLOS ONE. After careful consideration, we feel that it has merit but does not fully meet PLOS ONE’s publication criteria as it currently stands. Therefore, we invite you to submit a revised version of the manuscript that addresses the points raised during the review process.

Title: The Effect of Therapeutic Potential and Safety of Bone Marrow-Derived vs. Adipose-Derived Mesenchymal Stem Cells in Aged Mice with Septic Arthritis

This manuscript explores the comparative therapeutic efficacy and safety of bone marrow-derived versus adipose-derived mesenchymal stem cells in an aged murine model of septic arthritis. The topic is timely and clinically relevant, with strengths including the use of multiple outcome measures (histology, cytokine profiling, and safety assessment), as well as a focus on an aged population. However, there are significant concerns regarding clarity in experimental grouping (control vs. untreated) and replication. Some methodological details (differentiation assays, staining protocols, donor mice) are incomplete, and one referenced figure (Fig. 1) is missing. Addressing these issues will greatly improve clarity, reproducibility, and impact of the manuscript.

Reviewer Comments to Authors

1. Lines 111–112: The text states that only the treated groups were replicated. Please clarify what this means. Were the untreated and control groups not replicated? If so, why? Replication should ideally be consistent across all groups to avoid bias.

2. Line 120: It is stated that the untreated groups received no therapy. However, if all four untreated subgroups (Un-BM1, Un-BM2, Un-AD1, Un-AD2) only received PBS, the distinction between them is unclear. Why are they separately labeled as Un-BM1, Un-BM2, etc.? Please justify the rationale.

3. Line 125: Based on the description, the only difference is that the control group received PBS without arthritis induction, while the untreated groups received PBS after arthritis induction. If correct, please state this explicitly to avoid confusion.

4. Line 172: Please clarify whether separate donor mice were used for MSC isolation or whether experimental mice themselves were used as donors. If separate donors were used, how many animals were involved? This information is important for ethical and methodological transparency.

5. Line 219: The manuscript mentions differentiation but does not provide details of the differentiation assay. Please describe the protocols followed for osteogenic, chondrogenic, and adipogenic differentiation, including induction media and culture duration.

6. Lines 219–220: Only Alizarin Red staining for osteogenesis is mentioned. What staining protocols were used for adipogenesis (e.g., Oil Red O) and chondrogenesis (e.g., Alcian Blue or Safranin O)? Please include full details.

7. Line 222 (Figure 1): Figure 1 is referred to in the text but does not appear in the manuscript. Kindly ensure that all referenced figures are included.

8. Line 245: The manuscript states that arthritis was induced and MSCs were injected within 24 hours. How was arthritis confirmed to have developed before treatment administration? Were clinical, histological, or imaging assessments used, or was this timing based on prior references? Please clarify.

9. Lines 248–250: The description of untreated groups remains confusing here. Please clarify exactly what was administered and why these subgroups differ in labeling.

10. Lines 260–261: Please provide details of the protocols for Hematoxylin & Eosin, Safranin O, Masson’s Trichrome, COL2A1, and SOX9 staining/ IHC, including duration, concentrations, and source references.

11. Line 454: The phrase “untreated cell” is unclear. Does this mean cells were administered but not pre-treated, or does it refer to groups that received no MSCs at all? Kindly clarify terminology.

12. Line 687: The phrase should be revised to “used to” for grammatical accuracy.

13. Line 874: Please ensure that “BM-MSC” is abbreviated consistently at its first mention and uniformly used throughout the manuscript.

We look forward to receiving your revised manuscript.

Kind regards,

Md Shaifur Rahman, Ph.D

Academic Editor

PLOS ONE

Journal Requirements:

Additional Editor Comments:

Dear Authors,

Thank you for submitting your manuscript to PLOS ONE. We have now received feedback from our reviewers, and their comments are included at the bottom of this letter. The reviewers find the topic of your manuscript to be timely and relevant.

However, the reviewers have raised several concerns. We believe that addressing these points through careful revision will significantly strengthen the manuscript. Therefore, we invite you to submit a revised version of your manuscript that incorporates all of the reviewers' comments. Please note that your revisions should also include a full response to each point raised, detailing the changes made.

Given the nature of the requested revisions, we consider this decision to be an invitation for Minor Revision.

Please submit your revised manuscript sooner (e.g., 4-6 weeks). If you require additional time, please contact us.

We look forward to receiving your revised manuscript.

Sincerely,

Md Shaifur Rahman, PhD

Academic Editor

PLOS ONE

Title: The Effect of Therapeutic Potential and Safety of Bone Marrow-Derived vs. Adipose-Derived Mesenchymal Stem Cells in Aged Mice with Septic Arthritis

This manuscript explores the comparative therapeutic efficacy and safety of bone marrow-derived versus adipose-derived mesenchymal stem cells in an aged murine model of septic arthritis. The topic is timely and clinically relevant, with strengths including the use of multiple outcome measures (histology, cytokine profiling, and safety assessment), as well as a focus on an aged population. However, there are significant concerns regarding clarity in experimental grouping (control vs. untreated) and replication. Some methodological details (differentiation assays, staining protocols, donor mice) are incomplete, and one referenced figure (Fig. 1) is missing. Addressing these issues will greatly improve clarity, reproducibility, and impact of the manuscript.

Reviewer Comments to Authors

1. Lines 111–112: The text states that only the treated groups were replicated. Please clarify what this means. Were the untreated and control groups not replicated? If so, why? Replication should ideally be consistent across all groups to avoid bias.

2. Line 120: It is stated that the untreated groups received no therapy. However, if all four untreated subgroups (Un-BM1, Un-BM2, Un-AD1, Un-AD2) only received PBS, the distinction between them is unclear. Why are they separately labeled as Un-BM1, Un-BM2, etc.? Please justify the rationale.

3. Line 125: Based on the description, the only difference is that the control group received PBS without arthritis induction, while the untreated groups received PBS after arthritis induction. If correct, please state this explicitly to avoid confusion.

4. Line 172: Please clarify whether separate donor mice were used for MSC isolation or whether experimental mice themselves were used as donors. If separate donors were used, how many animals were involved? This information is important for ethical and methodological transparency.

5. Line 219: The manuscript mentions differentiation but does not provide details of the differentiation assay. Please describe the protocols followed for osteogenic, chondrogenic, and adipogenic differentiation, including induction media and culture duration.

6. Lines 219–220: Only Alizarin Red staining for osteogenesis is mentioned. What staining protocols were used for adipogenesis (e.g., Oil Red O) and chondrogenesis (e.g., Alcian Blue or Safranin O)? Please include full details.

7. Line 222 (Figure 1): Figure 1 is referred to in the text but does not appear in the manuscript. Kindly ensure that all referenced figures are included.

8. Line 245: The manuscript states that arthritis was induced and MSCs were injected within 24 hours. How was arthritis confirmed to have developed before treatment administration? Were clinical, histological, or imaging assessments used, or was this timing based on prior references? Please clarify.

9. Lines 248–250: The description of untreated groups remains confusing here. Please clarify exactly what was administered and why these subgroups differ in labeling.

10. Lines 260–261: Please provide details of the protocols for Hematoxylin & Eosin, Safranin O, Masson’s Trichrome, COL2A1, and SOX9 staining/ IHC, including duration, concentrations, and source references.

11. Line 454: The phrase “untreated cell” is unclear. Does this mean cells were administered but not pre-treated, or does it refer to groups that received no MSCs at all? Kindly clarify terminology.

12. Line 687: The phrase should be revised to “used to” for grammatical accuracy.

13. Line 874: Please ensure that “BM-MSC” is abbreviated consistently at its first mention and uniformly used throughout the manuscript.

Reviewers' comments:

Reviewer's Responses to Questions

**Comments to the Author**

Reviewer #2: All comments have been addressed

Reviewer #3: (No Response)

2. Is the manuscript technically sound, and do the data support the conclusions?

Reviewer #2: Yes

Reviewer #3: Partly

3. Has the statistical analysis been performed appropriately and rigorously?

Reviewer #2: Yes

Reviewer #3: I Don't Know

4. Have the authors made all data underlying the findings in their manuscript fully available?

Reviewer #2: Yes

Reviewer #3: Yes

5. Is the manuscript presented in an intelligible fashion and written in standard English?

Reviewer #2: Yes

Reviewer #3: Yes

Reviewer #2: The authors have adequately responded to all concerns raised in the previous round of review. The methodology is now clearly described, the data are appropriately presented, and the discussion has been strengthened to highlight the relevance and implications of the findings.

In my opinion, the manuscript is now suitable for acceptance in its current form.

Reviewer #3: Title: The Effect of Therapeutic Potential and Safety of Bone Marrow-Derived vs. Adipose-Derived Mesenchymal Stem Cells in Aged Mice with Septic Arthritis

This manuscript explores the comparative therapeutic efficacy and safety of bone marrow-derived versus adipose-derived mesenchymal stem cells in an aged murine model of septic arthritis. The topic is timely and clinically relevant, with strengths including the use of multiple outcome measures (histology, cytokine profiling, and safety assessment), as well as a focus on an aged population. However, there are significant concerns regarding clarity in experimental grouping (control vs. untreated) and replication. Some methodological details (differentiation assays, staining protocols, donor mice) are incomplete, and one referenced figure (Fig. 1) is missing. Addressing these issues will greatly improve clarity, reproducibility, and impact of the manuscript.

Reviewer Comments to Authors

1. Lines 111–112: The text states that only the treated groups were replicated. Please clarify what this means. Were the untreated and control groups not replicated? If so, why? Replication should ideally be consistent across all groups to avoid bias.

2. Line 120: It is stated that the untreated groups received no therapy. However, if all four untreated subgroups (Un-BM1, Un-BM2, Un-AD1, Un-AD2) only received PBS, the distinction between them is unclear. Why are they separately labeled as Un-BM1, Un-BM2, etc.? Please justify the rationale.

3. Line 125: Based on the description, the only difference is that the control group received PBS without arthritis induction, while the untreated groups received PBS after arthritis induction. If correct, please state this explicitly to avoid confusion.

4. Line 172: Please clarify whether separate donor mice were used for MSC isolation or whether experimental mice themselves were used as donors. If separate donors were used, how many animals were involved? This information is important for ethical and methodological transparency.

5. Line 219: The manuscript mentions differentiation but does not provide details of the differentiation assay. Please describe the protocols followed for osteogenic, chondrogenic, and adipogenic differentiation, including induction media and culture duration.

6. Lines 219–220: Only Alizarin Red staining for osteogenesis is mentioned. What staining protocols were used for adipogenesis (e.g., Oil Red O) and chondrogenesis (e.g., Alcian Blue or Safranin O)? Please include full details.

7. Line 222 (Figure 1): Figure 1 is referred to in the text but does not appear in the manuscript. Kindly ensure that all referenced figures are included.

8. Line 245: The manuscript states that arthritis was induced and MSCs were injected within 24 hours. How was arthritis confirmed to have developed before treatment administration? Were clinical, histological, or imaging assessments used, or was this timing based on prior references? Please clarify.

9. Lines 248–250: The description of untreated groups remains confusing here. Please clarify exactly what was administered and why these subgroups differ in labeling.

10. Lines 260–261: Please provide details of the protocols for Hematoxylin & Eosin, Safranin O, Masson’s Trichrome, COL2A1, and SOX9 staining/ IHC, including duration, concentrations, and source references.

11. Line 454: The phrase “untreated cell” is unclear. Does this mean cells were administered but not pre-treated, or does it refer to groups that received no MSCs at all? Kindly clarify terminology.

12. Line 687: The phrase should be revised to “used to” for grammatical accuracy.

13. Line 874: Please ensure that “BM-MSC” is abbreviated consistently at its first mention and uniformly used throughout the manuscript.

**Do you want your identity to be public for this peer review?** For information about this choice, including consent withdrawal, please see our Privacy Policy

Reviewer #2: No

Reviewer #3: No

---

## [Author Response · Author response to Decision Letter 3]

5 Oct 2025

Comments 1: Lines 111–112: The text states that only the treated groups were replicated. Please clarify what this means. Were the untreated and control groups not replicated? If so, why? Replication should ideally be consistent across all groups to avoid bias.

Response 1: The statement has been clarified. This is because each group was replicated four times. Please, refer to line 112-114 in the revised file.

Comments 2: Line 120: It is stated that the untreated groups received no therapy. However, if all four untreated subgroups (Un-BM1, Un-BM2, Un-AD1, Un-AD2) only received PBS, the distinction between them is unclear. Why are they separately labeled as Un-BM1, Un-BM2, etc.? Please justify the rationale.

Response 2: In line 110-112, Un-BM1 and Un-BM2 represent untreated bone marrow-derived MSCs (type 1 and 2, defined in the next paragraph), while Un-AD1 and Un-AD2 represent untreated adipose-derived MSCs (type 1 and 2). The only difference is that the control group received PBS without arthritis induction, while the untreated groups received PBS after arthritis induction. Please refer to lines 126-128.

Comments 3: Line 125: Based on the description, the only difference is that the control group received PBS without arthritis induction, while the untreated groups received PBS after arthritis induction. If correct, please state this explicitly to avoid confusion.

Response 3: Yes, that is correct. This clarification has been added to the revised file. We appreciate it.

Comments 4: Line 172: Please clarify whether separate donor mice were used for MSC isolation or whether experimental mice themselves were used as donors. If separate donors were used, how many animals were involved? This information is important for ethical and methodological transparency.

Response 4: A separate mice were used as donor. The detail information regarding this has been added (lines 76-89). Please refer to revised file.

Comments 5: Line 219: The manuscript mentions differentiation but does not provide details of the differentiation assay. Please describe the protocols followed for osteogenic, chondrogenic, and adipogenic differentiation, including induction media and culture duration.

Response 5: The detail protocols have been provided in the revised file in lines 242-277.

Comments 6: Lines 219–220: Only Alizarin Red staining for osteogenesis is mentioned. What staining protocols were used for adipogenesis (e.g., Oil Red O) and chondrogenesis (e.g., Alcian Blue or Safranin O)? Please include full details.

Response 6: The detail protocols have been provided in lines 242-277.

Comments 7: Line 222 (Figure 1): Figure 1 is referred to in the text but does not appear in the manuscript. Kindly ensure that all referenced figures are included.

Response 7: Something must have happened in the system but attached all figures at submission. However, we have re-attached Figure 1 in this revision.

Comments 8: Line 245: The manuscript states that arthritis was induced and MSCs were injected within 24 hours. How was arthritis confirmed to have developed before treatment administration? Were clinical, histological, or imaging assessments used, or was this timing based on prior references? Please clarify.

Response 8: We have clarified and provide detail how we confirmed the development of arthritis before treatment administration. Please, refer to 352-364.

Comments 9: Lines 248–250: The description of untreated groups remains confusing here. Please clarify exactly what was administered and why these subgroups differ in labeling.

Response 9: The statement has been revised accordingly for more clarity. Please refer to lines 347-349.

Comments 10: Lines 260–261: Please provide details of the protocols for Hematoxylin & Eosin, Safranin O, Masson’s Trichrome, COL2A1, and SOX9 staining/ IHC, including duration, concentrations, and source references.

Response 10: The detail protocols have been provided in lines 280-324. Please refer to page 12-14.

Comments 11: Line 454: The phrase “untreated cell” is unclear. Does this mean cells were administered but not pre-treated, or does it refer to groups that received no MSCs at all? Kindly clarify terminology.

Response 11: The statement has been revised by removing the confusing phrase. Please refer to line 564.

Comments 12: Line 687: The phrase should be revised to “used to” for grammatical accuracy.

Response 12: It has been revised as suggested.

Comments 13: Line 874: Please ensure that “BM-MSC” is abbreviated consistently at its first mention and uniformly used throughout the manuscript.

Response 13: It has been revised accordingly.

We thank the reviewer for time and effort in helping us improve our work.

---

## [Editor Report · Decision Letter 3]

6 Oct 2025

The Effect of Therapeutic Potential and Safety of Bone Marrow-Derived against Adipose-Derived Mesenchymal Stem Cells in Aged Mice Associated with Septic Arthritis

PONE-D-25-03442R3

Dear Dr. Khalid Ahmed,

We’re pleased to inform you that your manuscript has been judged scientifically suitable for publication and will be formally accepted for publication once it meets all outstanding technical requirements.

Kind regards,

Md Shaifur Rahman, Ph.D

Academic Editor

PLOS ONE
---

## [Editor Report · Acceptance letter]

PONE-D-25-03442R3

PLOS ONE

Dear Dr. Khalid Ahmed,

I'm pleased to inform you that your manuscript has been deemed suitable for publication in PLOS ONE. Congratulations! Your manuscript is now being handed over to our production team.

Kind regards,

on behalf of

Dr. Md Shaifur Rahman

Academic Editor

PLOS ONE